

**Land-atmosphere interactions in sub-polar and alpine climates in the CORDEX FPS**
**LUCAS models: I. Evaluation of the snow-albedo effect**
Anne Sophie Daloz[1], Clemens Schwingshackl[1,13], Priscilla Mooney[2], Susanna Strada[3], Diana Rechid[4],
Edouard L. Davin[5], Eleni Katragkou[6], Nathalie de Noblet-Ducoudré[7], Michal Belda[8], Tomas Halenka[8],
Marcus Breil[9], Rita M. Cardoso[10], Peter Hoffmann[4], Daniela C.A. Lima[10], Ronny Meier[5], Pedro M.M.
Soares[10], Giannis Sofiadis[6], Gustav Strandberg[11], Merja H. Toelle[12] and Marianne T. Lund[1].
1. CICERO Center for International Climate Research, Oslo, Norway
2. NORCE Norwegian Research Centre, Bjerknes Centre for Climate Research, Bergen, Norway
3. International Center for Theoretical Physics, Trieste, Italy
4. Climate Service Center Germany, Helmholtz-Zentrum Hereon, Hamburg, Germany
5. Wyss Academy for Nature, Climate and Environmental Physics, Oeschger Center for Climate Change Research,
University of Bern, Bern, Switzerland
6. Department of Meteorology and Climatology, School of Geology, Aristotle University of Thessaloniki,
Thessaloniki, Greece
7. Laboratoire des Sciences du Climat et de l'environnement, Paris, France
8. Department of Atmospheric Physics, Faculty of Mathematics and Physics, Charles University, Prague, Czech
Republic
9. Institute for Meteorology and Climate Research, Karlsruhe Institute of Technology, Karlsruhe, Germany
10. Instituto Dom Luiz, Faculdade de Ciências da Universidade de Lisboa, 1749-016 Lisboa, Portugal
11. Swedish Meteorological and Hydrological Institute, Norrkoping, Sweden
12. Center for Environmental Systems Research, University of Kassel, Germany
13. Department of Geography, Ludwig-Maximilians-Universität, Munich, Germany.
*Corresponding author: Anne Sophie Daloz (anne.sophie.daloz@cicero.oslo.no)*



**Abstract**
In the Northern Hemisphere, the seasonal snow cover plays a major role in the climate system via its
effect on surface albedo and fluxes. The parameterization of snow-atmosphere interactions in climate
models remains a source of uncertainty and biases in the representation of the local and global climate.
Here, we evaluate the ability of an ensemble of regional climate models (RCMs) coupled to different
land surface models to simulate the snow albedo effect over Europe, in winter and spring. We use a
previously defined index, the Snow Albedo Sensitivity Index (SASI), to quantify the radiative forcing
due to the snow albedo effect. By comparing RCM-derived SASI values with SASI calculated from
reanalyses and satellite retrievals, we show that an accurate simulation of snow cover is essential for
correctly reproducing the observed forcing over mid- and high-latitudes in Europe. The choice of
parameterizations with first and foremost the choice of the land surface model but also the convection
scheme and the planetary boundary layer, strongly influences the representation of SASI as it affects the
ability of climate models to simulate snow cover correctly. The agreement between the datasets differs
between the accumulation and ablation periods, with the latter one presenting the greatest challenge for
the RCMs. Given the dominant role of land surface processes in the simulation of snow cover during
the ablation period, the results suggest that the choice of the land surface model is more critical for the
representation of SASI than the atmospheric model during this time period.





## 1. Introduction

Snow is an important part of the climate system as it regulates the temperature of the Earth's surface via its effect on surface albedo and surface fluxes. In mid- and high-latitude regions, snow is the main interface through which land interacts with the atmosphere during the cold season and the importance of snow-atmosphere interactions in modulating the energy budget at high latitudes during winter has been demonstrated (Diro and Sushama, 2018; Henderson et al., 2018; Xu and Dirmeyer, 2013). Snow cover extent and depth can modify both surface energy and moisture budgets, triggering complex feedback mechanisms that impact both local and remote climates (Diro and Sushama, 2018). In particular, snow can have a strong impact on climate due to its high albedo, primarily because of the contrast in the surface energy balance between snow-covered and snow-free land surfaces (Qu and Hall, 2014). Reciprocally, with climate change, rising temperatures are already altering the Earth's snow amount and occurrences, for example shortening the snow season in Eurasia (Ye and Cohen, 2013; Gobiet et al., 2014; Mioduszewski et al., 2015; Beniston et al., 2018; Matiu et al., 2020). In this context, it is crucial to better understand snow-atmosphere processes and the ability of climate models to represent them.

The direct impact of snow on the atmosphere is known as the snow albedo effect (SAE; Xu and Dirmeyer, 2011, 2013), where the presence of snow affects the land surface energy budget and influences the local climate, modifying air temperature. To quantify the contribution from the SAE to the snow-atmosphere coupling, Xu and Dirmeyer (2011) developed the Snow Albedo Sensitivity Index (SASI). This index combines incoming shortwave radiation with snow cover variability to quantify the snow-albedo coupling strength, i.e. SASI estimates the degree to which the atmosphere responds to anomalies in snow cover. Applying SASI to satellite observations, Xu and Dirmeyer (2011) found that the coupling between snow and albedo is particularly strong during the snowmelt period in the Northern Hemisphere. At high-latitudes, for example, the effects of snow cover on the climate is strongly related to the way vegetation cover is prescribed. Removal of boreal forests locally reduces surface air temperature and precipitation by increasing surface albedo and decreasing plant evapotranspiration. The strength of the coupling between snow and the atmosphere is determined by processes involving



radiative fluxes but also hydrology. Therefore, Xu and Dirmeyer (2013) also defined the snow
hydrological effect (SHE), which is a result of soil moisture anomalies from snowmelt. Through land-
atmosphere interactions, they have a delayed impact on the atmosphere. Besides these direct and indirect
effects, positive and negative snow-atmosphere feedbacks, such as the snow-albedo feedback (SAF; Qu
and Hall, 2007; Fletcher et al., 2015; Thackeray et al., 2018) can amplify anomalies. The SAF represents
changes in surface albedo from cooling (warming) that can cause decreases (increases) in absorbed solar
radiation, amplifying the initial cooling (warming). It is an important driver for regional climate change
in Northern Hemisphere land areas.

Here, we investigate the ability of an ensemble of RCMs to represent snow cover and the

radiative forcing from the snow albedo effect (SASI) over Europe, including a comparison between mid-
and high-latitude regions. We derive SASI using radiative fluxes and snow cover from satellites,
reanalysis and model outputs. Building on findings by Xu and Dirmeyer (2011, 2013), we focus on
winter and spring seasons, i.e. transitioning from the accumulation to the ablation period, when SASI is
reaching a maximum. While some previous studies have investigated snow-atmosphere processes in
climate models for specific regions (e.g. European Alps; Magnusson et al., 2010; Matiu et al., 2019;
Lüthi et al., 2019), the literature remains limited. Here, we use the RCMs outputs from the flagship pilot
study Land Use and Climate Across Scale (LUCAS; Rechid et al., 2017; Breil et al., 2020; Davin et al.,
2020; Reinhart et al., 2020; Sofiadis et al., 2021). It is endorsed by the Coordinated Regional Climate
Downscaling Experiment (CORDEX) of the World Climate Research Programme (WCRP) over the
European domain (EURO-CORDEX, Jacob et al., 2020) and it enables us to perform a broader
assessment of several RCMs within a consistent framework. Our assessment is carried out in two parts
and published in companion articles. In Part I, we investigate the ability of these RCMs to represent the
SASI under present-day land cover distribution, while in Part II we explore the effects of large-scale
changes in vegetation cover. In LUCAS, each RCM performed three coupled land-atmosphere
experiments at the European scale: two idealized and intensive land use change experiments (GRASS
and FOREST) and a control experiment (EVAL). The GRASS and FOREST experiments will be



examined in the companion paper (Part II) while here, we use ten models from the EVAL experiment
only, which employ their standard land use and land cover maps.
Section 2 introduces the modeling and observational datasets used in this study as well as the
derivation of SASI, while Section 3 examines and discusses the ability of climate models to represent
SASI compared with satellite observations and reanalyses, focusing on the strength and timing of the
signal. Further, the origin of the differences between the models are explored by evaluating potential
common biases in the ensemble of simulations as well as individual model biases. The analysis also
explores the differences in SASI between mid- and high-latitude regions, opening the discussion on the
impacts of different land cover for the simulation of SASI, which will be further explored in Part II.
Finally, Section 4 the last sections offer some concluding remarks.
**2. Data and methodology**
**2.1 LUCAS experiments and models**
**2.1.1 The LUCAS experiments**
The simulations from the flagship pilot study LUCAS simulations cover the standard EURO-
CORDEX domain (Jacob et al., 2014) with a horizontal grid resolution of 0.44° (around 50 km). All
RCMs in LUCAS use a rotated coordinate system except the RegCM model, which applies a Lambert
conformal projection (suitable for mid-latitudes) on a regular grid. Here we use outputs from the EVAL
experiment, which employ land use and land cover maps; the GRASS and FOREST experiments will
be examined in the companion paper (part II). All simulations span the period 1986–2015 (with a spin-
up period ranging from one up to six years depending on the model) and take lateral and boundary
conditions from the ERA-Interim reanalysis (Dee et al., 2011). More details can be found in Davin et
al. (2020).
**2.1.2 Models and configurations**
We use outputs from ten coupled surface-atmosphere RCM simulations that participated in the
LUCAS project. The main model characteristics that are important for snow albedo coupling are
summarized in Table 1. The model ensemble presents five different RCMs: COSMO-CLM version 5.0-
clm9 (Sørland et al., 2021), WRF version 3.8.1 (Skamarock et al., 2008), RegCM versions 4.6 and 4.7





(Giorgi et al., 2012), RCA4 (Strandberg et al., 2015) and REMO (Jacob et al., 2012). These RCMs
contributed with different setups and configurations as described in Table 1. For example, the same
RCM is coupled to different land surface models (LSMs): COSMO-CLM is coupled to three distinct
LSMs, which are CLM5.0 (Lawrence et al., 2020), VEG3D (Breil and Schadler, 2017) and TERRA-
ML (Schrodin and Heise, 2002). WRF is coupled with either CLM4.0 (Oleson et al., 2010) or NOAH-
MP (Niu et al., 2011). Vice versa, the same LSM is combined with different versions of RCMs. The
CLM4.5 (Oleson et al., 2013) LSM is coupled to two distinct versions of RegCM (4.6 and 4.7) which
also differ in their choice of convection schemes. There are also two institutes with the same RCM and
LSM (WRF and Noah-MP) but different parameterizations, as they use distinct planetary boundary layer
(PBL) schemes. A detailed description of the RCMs is provided by Davin et al. (2020). For the analyses
in the present study, we use daily and monthly model outputs for incoming shortwave radiation and
snow cover. For deriving SASI, the native grid of the models was kept, minimising data loss. The other
fields were interpolated to a common 0.5°x0.5° grid using Climate Data Operators (CDO) bilinear
remapping.
**2.1.3 Snow-buried fraction of vegetation in models**

At high-latitudes, the effects of snow cover on regional climate strongly depend on the

prescribed vegetation cover. Removal of boreal forests locally reduces surface air temperature and
precipitation by increasing surface albedo and the duration of the snow cover and by decreasing plant
evapotranspiration. Today, the role of forest albedo on winter-spring climate in the high-latitudes is well
acknowledged based on field campaigns such as the Boreal Ecosystem-Atmosphere Study (BOREAS;
Betts et al., 2001) and on modeling studies (e.g., Betts and Ball, 1997; Betts et al., 1996; Betts et al.,
2001; Bonan, 2008; Davin and Noblet-Ducoudré, 2010; Mooney et al., 2021). These studies led to
implementing more sophisticated snow sub-models in LSMs that account for the burial of vegetation by
snow cover.

In the LUCAS ensemble, all LSMs, except for the TERRA-ML LSM used by CCLM-TERRA,

adjust the effective Leaf and Stem Area Index for snow-buried vegetation by adopting similar
approaches. Being a bulk/one-dimensional LSM, TERRA-ML applies an infinitesimal vegetation layer





on top of the soil surface and has no canopy (i.e., vegetation lays flat on the surface). However, to
correctly simulate the effect of trees masking the ground snow on radiation, TERRA-ML applies a
reduction factor for the snow albedo when vegetation such as forest canopies masks the snow. Hence,
when vegetation is snow-buried, all LSMs account for a highly reflecting surface in the calculation of
surface albedo. In Table 1, interested readers may find references to RCM-dependent snow-buried
vegetation schemes.
In terms of snow schemes, some LSMs contain more sophistication than others. Compared to
previous CLM versions (i.e., CLM4.0 and CLM4.5), CLM5.0 used by CCLM-CLM5.0 counts more
snow layers (12 instead of 5), treats separately canopy intercepted snow and more realistically captures
temperature and wind effects on the density of fresh snow (Lawrence et al., 2020; van Kampenhout et
al., 2017). The RCA4 model system and its internal LSM, used in RCA, include sub-grid orography in
the snow cover to capture inhomogeneous snow cover in mountainous areas. Noah-MP allows for 3
snow layers, depending on the total snow depth. To provide a better representation of the ground heat
fluxes, the first very layer is only 0.045 m thick. Noah-MP also considers snow interception by the
canopy, accounting for wind and temperature effects on snow accumulation and precipitation from the
canopy, snow melting and refreezing (Niu and Yang, 2004). The ground snow cover fraction is a
function of the snow depth and density and ground roughness (Niu et al., 2007)
**2.2 Reanalyses and remote sensing data**
Reanalysis data from ERA5-Land (Muñoz Sabater, 2019; Muñoz Sabater et al., 2021) and
MERRA-2 (Gelaro et al., 2017) as well as satellite data from the Moderate Resolution Imaging
Spectroradiometer (MODIS; Hall and Riggs, 2016) are used to evaluate the modelled snow distribution
and radiation in the RCMs. Specifically, we use monthly data for snow cover (variable "fractional area
of land snow cover" in MERRA-2), incoming shortwave radiation from ERA5-Land and MERRA-2,
and daily snow cover data from the MODIS sensors AQUA and TERRA. The reanalysis data are
interpolated bilinearly to the common 0.5°x0.5° grid (see Section 2.2). Reanalysis data cover the time
period 1986-2015 and MODIS data the period 2003-2015.
For MODIS data, the following processing steps are applied:





1. Data are masked according to the prevailing cloud cover since high cloud cover prevents a correct estimation of snow cover. We apply two different thresholds (20% and 50%) to the percent of clouds in each cell.

2. Only data flagged as "best", "good", and "ok" are used while all other data are masked.

3. Data are conservatively remapped to the common 0.5°x0.5° grid. Conservative remapping is chosen due to the large difference in resolution between the original MODIS data (0.05°) and the target grid (0.5°). It considers all grid points in the interpolation while, e.g., bilinear interpolation would only consider the neighbouring grid cells of the target grid.

4. A land-sea mask is applied to make sure that only land grid points are included in the analysis. Only grid points with more than 50% land fraction are included.

The masking for MODIS data implies that single grid points can contribute differently to the average over one region. To make the models and reanalyses comparable, each grid point is weighted by the amount of available MODIS data (individually for each month of the whole time period).

**2.3 Snow Albedo Sensitivity Index (SASI) and geographical scope**

SASI is an index that quantifies the climate forcing due to the snow albedo effect (Xu and Dirmeyer, 2013). It is defined as:

$$SASI = SW * \sigma(f_{sno})\Delta\alpha \qquad (1)$$

where $SW$ is the net shortwave radiation at the surface, $\sigma(f_{sno})$ is the standard deviation of snow cover fraction, and $\Delta\alpha$ is the average difference between the albedo of a snow-covered surface and the albedo of a snow-free surface. $\Delta\alpha$ is a constant value of 0.4 as assumed in Xu and Dirmeyer (2013). SASI is in $Wm^{-2}$ and high values of SASI, such as 10 $Wm^{-2}$, indicate a strong climate forcing from the snow albedo effect (Xu and Dirmeyer, 2013).

To better understand geographical differences in the role of snow for land-atmosphere coupling, we focus on three sub-regions over Europe, with different climate, vegetation cover, topography or latitudes: Scandinavia [5ºE-30ºE, 55ºN-70ºN], East Europe [16 ºE-30ºE, 44ºN-55ºN] and East Baltic [20ºE-40ºE, 50ºN-62ºN] (see Figure 1). The first two regions, Scandinavia and East Europe correspond





to regions 8 and 5 of the PRUDENCE project (Prediction of Regional scenarios and Uncertainties for
Defining EuropeaN Climate change risk and Effects; Christensen and Christensen, 2007). The three
selected regions differ in terms of climate but also in terms of vegetation: vegetation in Scandinavia is
mostly trees while the two other regions are covered by cropland and trees. The Scandinavian region
also stands out because of its geographical location covering high latitudes, where the incoming
shortwave radiation is very small or zero during winter. In comparison with the East Baltic region, which
is covered by plains, the East Europe and Scandinavia regions have a more complex topography as they
encompass the Carpathian and Scandinavian mountains, respectively.

**3. Results and discussion**
**3.1 SASI in satellite observations, reanalyses and RCMs over Europe**

In Figure 2, we first show the geographical distribution of SASI over Europe based on satellite

observations, the ERA5-Land reanalysis and the LUCAS models from January to June, averaged over
the 1986-2015 period. Focusing first on the satellite observations and ERA5-Land, an increase in SASI
can be observed during the first months of the year when solar radiation increases and snow is
accumulating (accumulation period), reaching a maximum in March or April depending on the region
examined, and then decreasing when snow starts melting (ablation period). At higher latitudes snow
melts later than at mid-latitudes, giving rise to SASI values during spring, as shown in Figure 2. Then,
SASI reaches very low values in May and June when the snow has melted almost entirely. This is as
expected, and the overall seasonal trend is consistent with Xu and Dirmeyer (2013). The model data
exhibits the same overall spatiotemporal cycle in SASI as the satellite observations and ERA5-Land.
However, large differences can be seen between the simulations in terms of amplitude or pattern,
especially during the ablation period. In March over the Carpathian Mountains, for example, SASI varies
between 1 $Wm^{-2}$ for WRFa-NoahMP and RCA, and 10 $Wm^{-2}$ for CCLM-CLM5.0 and RegCMa-
CLM4.5. It is also noteworthy that for almost all the models, SASI is close to zero everywhere in
continental Europe in May and June, as the snow has almost entirely melted, while in May for RegCMb-
CLM4.5 and CCLM-VEG3D there are still high values of SASI (~10 $Wm^{-2}$).



The ensemble of simulations run for LUCAS enable us to discuss the role of different
components of the RCMs, such as the land and atmosphere models or the choice in parameterizations.
For example, WRFc-NoahMP and WRFa-NoahMP show noticeable differences in the amplitude and
pattern of SASI (Fig. 2), even though they use the same LSM (Noah-MP) and atmospheric model
(WRF). Their differences come from parameterizations (planetary boundary layer and convection), thus
demonstrating the importance of atmospheric processes and their model representation for representing
snow processes. Then, WRF configuration coupled with the LSM CLM4.0 (WRFb-CLM4.0) also shows
different results from when it is coupled with NOAH-MP. For example, WRFa-NoahMP shows an
earlier poleward migration of high SASI values compared to WRFb-CLM4.0, moving north about one
month before WRFb-CLM4.0. Large differences can also be observed between CCLM-CLM5.0,
CCLM-TERRA, and CCLM-VEG3D; they all use the same RCM but different LSMs. In contrast to the
two other Cosmo configurations, CCLM-VEG3D uses a snow flag for snow cover (i.e., indicates if snow
is present or not; Section 2.3), explaining its different representation of SASI. This suggests that SASI
is very sensitive to the configurations of and process parameterizations in the climate model. In
particular, the choice of the LSM or certain parameterizations (e.g. convection scheme) highly influence
the representation of the climate forcing from the snow albedo effect. The role of the LSM in this context
will be investigated further in the coming sections of the article.

**3.2 Transition between the accumulation and ablation periods**

To further investigate the differences in snow albedo coupling between the simulations and the
observation-based datasets during the accumulation and ablation periods, a time-series of SASI from
January to June is presented in Figure 3 for the three sub-regions East Europe, East Baltic and
Scandinavia (see Figure 1 for their extents). Before looking at the differences between the different
datasets, it is interesting to compare the amplitude of SASI between East Baltic and East Europe (mid-
latitude regions) with Scandinavia (high-latitude region), which shows slightly higher values of SASI
over the mid-latitude regions in satellite observations, ERA5 and most of the RCMs. This confirms
previous findings from Xu and Dirmeyer (2013), which estimated higher values of SASI in mid- versus



high-latitude regions in satellite observations. However, even with higher values at mid-latitudes, this
result suggests that the radiative forcing due to the snow albedo effect is not negligible over high-latitude
regions in winter and spring. This result shows again the importance of the snow-atmosphere processes
in mid- and high-latitudes in the Northern hemisphere.

Then, coming back to the comparison of the different datasets, in all three regions, the models

and observations indicate a pronounced peak in SASI. The maximum in SASI marks the transition
between the accumulation and ablation periods. The timing of this transition depends on the region
examined due to, for example, latitudinal differences in incoming solar radiation. Although the
amplitude of the peak is very similar between the satellite observations and ERA5-Land, it is interesting
to see that the timing differs between them, over Scandinavia and East Baltic. Over East Europe it
happens in March for both the satellite observations and ERA5, for East Baltic in March (satellites) or
April (ERA5) and for Scandinavia in April (satellites) or May (ERA5). The origin of these differences
has not been clarified yet. This might be due to the higher elevations of these two regions compared to
East Europe as complex orography is a driving factor for the spatial heterogeneity of precipitation
(Grunewald et al., 2014).

The LUCAS simulations also show a pronounced peak in SASI in all regions (Fig. 3), however

they do not all agree on the timing and the amplitude of the signal. For example, in the East Baltic
region, some models (WRFc-NoahMP and WRFa-NoahMP) simulate a peak in March, others in April
(WRFb-CLM4.0 and CCLM-CLM5.0) or even in May (RegCMb-CLM4.5 and CCLM-VEG3D). In
general, RegCMb-CLM4.5 and CCLM-VEG3D tend to present the latest peak in SASI as well as the
highest amplitude in the signal. On the other hand, WRFa-NoahMP tends to produce an earlier peak and
lower values of SASI, especially over East Europe. These differences might be related to the way snow
melts in the different models and will be further explored in the next section. More generally, we see
that during the accumulation period, all the datasets are in better agreement compared to the ablation
period (Fig. 3). For East Europe and East Baltic, the spread largely increases in March and for
Scandinavia from April until the end of the season, when the snow is melting.

This large model spread during the ablation period is further confirmed by Figure 4 showing the

pattern correlation between the simulations and ERA5-Land from January to June. For many models,



the correlation is high at the beginning of the season but strongly decreases in March or April, when the
snow starts melting. These results are in agreement with previous studies showing the difficulties of
climate models to represent snow processes during the ablation period (Essery et al. 2009). Given the
dominant role of land surface processes during the ablation period, this suggests that the choice of the
LSM is more critical for the representation of the climate forcing from the snow albedo effect than the
atmospheric model in spring. For calculating snow-covered areas at different stages of ablation, a correct
representation of the landscape type is important (Pomeroy et al., 1998). Figure 4 also shows that the
behavior of the RCMs is different between East Europe and East Baltic versus Scandinavia. Over the
latter region, most RCMs differ from the reanalysis indicated by low correlations. Earlier studies showed
that snow accumulates or melts very differently in an open region compared to a forested region (Jonas
and Essery, 2014; Moeser et al., 2016). Our results suggest that RCMs represent snow processes better
in open spaces like the East Baltic than in forest-covered regions like Scandinavia. The relationship
between the representation of SASI and land cover will be further explored in the companion article,
Part II. The mountains in Scandinavia could also be a source of biases since the resolution of the RCM
simulations (0.44°) can be considered insufficient to represent the more complex topography of
Scandinavia.

**3.3 Inter-model differences in SASI**
To better understand the origin of the differences in SASI across RCMs, we explore the
relationship between SASI and its components, surface snow cover and shortwave radiation, during the
accumulation and ablation periods. Figure 5 presents a comparison of the averaged monthly surface
snow cover for the LUCAS simulations, the reanalyses MERRA-2 and ERA5-Land as well as the
satellite observations from MODIS, averaged over our three regions of interest, from January to May.
First, it should be noted that differences can be observed between the reanalyses and the satellite
observations as the different datasets have their own limitations or biases. For example, the surface snow
cover in East Baltic in March is ~0.6 for MODIS, ~0.7 for MERRA-2 and ~0.8 for ERA5-Land. It is
therefore important to include several observation-based datasets to evaluate the ability of climate
models to represent snow cover and estimate the uncertainties associated with this variable. The



representation of snow cover in RCMs can also be different depending on the model examined. Over
Scandinavia, snow cover varies between 0.4 for WRFa-NoahMP and 1.0 for WRFb-CLM4.0 in January.
For the same month, the differences are even higher for the other two regions, varying between 0.3 for
WRFa-NoahMP and 1.0 for WRFb-CLM4.0 in East Baltic, and 0.1 for WRFa-NoahMP and 1.0 for
WRFb-CLM4.0 in East Europe. Although there are already differences during the accumulation period,
Figure 5 shows that the spread increases when the snow starts melting. This result indicates a common
bias between the models that highly disagree with the reanalysis and observations, regarding snow cover
in spring. This confirms the result from the previous section as it is again pointing towards a bias from
LSMs as this part of the RCM is primordial for representing land surface processes related to snow cover
during the ablation period.

Based on Figure 3, RegCMb-CLM4.5 and CCLM-VEG3D were identified as models with

higher values in SASI during the ablation period and later peaks for all regions. Figure 5 shows that this
behavior can be at least partly attributed to their representation of snow cover. During the ablation
period, they all tend to produce higher values of snow cover compared to the other models and also to
keep high values later in the season. This behavior is confirmed by the black dots under these two models
during the ablation period as they indicate when the models are outside the range of the reference
datasets (MERRA-2, ERA5-Land and MODIS). This is particularly striking for CCLM-VEG3D.
Similarly, the low SASI peaks for WRFa-NoahMP, which also occur earlier than the peaks for other
models (Figure 3), might be related to the lower values in snow cover and the small interannual snow
cover variability compared to the other RCMs, particularly in East Europe (Figure 5). Again, this is
confirmed by the black dots indicated under the model. The differences in snow cover are also reflected
by the rate of snow melting for the different RCMs (Supplemental Material; Figure S1). The models
having high snow cover late in spring (RegCMb-CLM4.5 and CCLM-VEG3D) tend to have later snow
melt than the other models while WRFa-NoahMP, showing reduced snow cover earlier than the other
models, tends to melt sooner.

Another component of SASI is shortwave radiation at the surface, which is presented in Figure

6 for the LUCAS simulations, the reanalyses MERRA-2 and ERA5-Land, averaged over our three
regions of interest, from January to May. The comparison between the RCMs and the reanalysis shows





noticeable differences for some models. Both REMO-iMOVE and CCLM-VEG3D exhibit very
different results in terms of surface shortwave radiation compared to the datasets as shown by the black
dots on the figure, showing much lower and higher values, respectively. However, even with these
discrepancies, they both reproduce SASI reasonably well. This seems to indicate that the differences in
the representation of the forcing from the snow albedo effect are mostly driven by differences in the
representation of snow cover in the models. This is confirmed by Figure 7 showing the average
correlation across models between SASI and shortwave radiation (left) as well as SASI and snow cover
(right) for the LUCAS models. Scandinavia and East Baltic present similar results with significant,
positive correlations between SASI and snow cover for almost all months, associated with positive but
not significant correlations between SASI and shortwave radiation. For East Europe, the correlation
between SASI and snow cover is low and not significant in January and February but remains high and
significant the rest of the time period. In parallel, the correlation between SASI and downward
shortwave radiation at the surface is negative for almost all months and not significant. Overall, high
and significant correlations often appear between SASI and snow cover for the three regions from
January to June. On the other hand, the correlations between SASI and shortwave radiation are low and
usually not significant. This indicates that the differences in the representation of the forcing from the
snow albedo effect are mostly driven by differences in the representation of snow cover in the models.

**4. Conclusion**

Previous work already showed the difficulty for climate models to represent snow variables or

processes, such as snow cover and depth (Matiu et al., 2020) or the SAF (Fletcher et al., 2015), however
the origin of the differences between the models is not clear yet. In this work, we focus on the ability of
RCMs to simulate the radiative forcing from the snow albedo effect in winter and spring over Europe
and explore the origin of the differences between the RCMs. This forcing is represented by the index
SASI, which quantifies the strength of the coupling between snow and albedo. Ten RCMs from the
CORDEX Flagship Pilot Study LUCAS are compared to satellite observations and reanalysis including
ERA5-Land and MERRA-2. These simulations are part of the control experiment of LUCAS, which



uses the standard EURO-CORDEX domain (Jacob et al., 2014) with a horizontal grid resolution of 0.44°
(around 50 km).

The results show that climate models are able to reproduce some of the SASI characteristics

(e.g. existence of a peak, amplitude of the peak) compared to reanalysis and satellite observations
(Section 3.1), even if large differences appear between the RCMs. The climate models' ability to
represent SASI is highly related to their representation of snow cover (Section 3.3), which can be
difficult to represent for climate models (Matiu et al., 2020). Our results also suggest that the models'
capability highly differs between the accumulation and ablation periods. Most models have much lower
agreement with reanalyses and satellite observations in the ablation period, with some exceptions (e.g.
CCLM-CLM5.0 over East Europe), indicating a common bias regarding snow cover in spring, pointing
towards a bias from LSMs. This bias seems to be common to most LSMs even if they are based on
different assumptions and parameterizations (see Section 2.3). It is also interesting that even though
CCLM-TERRA is not as advanced in terms of snow modeling compared to the other models (e.g.
Section 2.1.3), it still manages to represent SASI reasonably well over Europe. In addition, the
representation of the sub-grid scale surface heterogeneity (Table 1; PFT-dominant versus PFT-tile) does
not seem to affect the ability of RCMs to represent snow cover or SASI.

Although it is difficult to identify the origin of the bias in the RCMs, an increase in spatial

resolution might improve the simulation of snow cover and therefore the representation of SASI. For
example, over Scandinavia, an increase in spatial resolution would provide a better representation of the
complex topography of the region as well as its forested areas, which may lead to an improved
simulation of the coupling between snow and albedo. The coming phases of LUCAS, phases 2 and 3,
could help answer this question as they will produce simulations at a higher spatial resolution, 12 km
and convection-permitting (<3km) respectively. Taking advantage of the different configurations of the
LUCAS simulations, we have also explored the role of distinct parts of the models in their ability to
represent SASI. The first part of this work has already emphasized the role of the LSMs, but other
components can also play an important role. WRFc-NoahMP and WRFa-NoahMP, even though using
the same RCM and LSM, show noticeable differences in the amplitude and pattern of SASI. Their



differences in parameterizations (planetary boundary layer and convection) are certainly affecting the

way they represent SASI, highlighting the impact of such choices and the role of atmospheric processes.

Mid- and high-latitude areas are also specifically examined looking at three sub-regions:

Scandinavia, East Europe and East Baltic (Section 3.2). The comparison of the three sub-regions shows

the difficulties for models to simulate SASI over Scandinavia during the accumulation and ablation

periods. The simulation of snow processes in a forested region is more challenging than in an open

region (Jonas and Essery, 2014; Moeser et al., 2016). Thus, potentially climate models can have more

difficulties representing snow processes in forest-covered regions like Scandinavia compared to open-

land regions like East Baltic. The relationship between the representation of SASI and land-cover will

be further explored in the companion article (Part II), analyzing the other experiments (GRASS and

FOREST) from LUCAS. Finally, the comparison of mid- versus high-latitude regions shows slightly

higher values of SASI over the mid-latitude regions in satellite observations, ERA5 and most of the

RCMs. This confirms previous findings from Xu and Dirmeyer (2013), which estimated higher values

of SASI in mid- versus high-latitude regions in satellite observations. Our results also suggest that the

climate forcing due to the snow albedo effect is not negligible over high-latitude regions in winter and

spring. This is important since often the land-atmosphere coupling is considered weaker at higher

latitudes (Xu and Dirmeyer, 2011) but it is also possible that this coupling happens through snow and is

therefore underestimated.

**Acknowledgements**

CICERO researchers acknowledge funding from the Norwegian Research Council (grant 254966). In

Norway, the simulations were stored on the server NIRD with resources provided by UNINETT Sigma2

- the National Infrastructure for High Performance Computing and Data Storage in Norway. WRFc-

NoahMP simulations were performed and stored on resources provided by UNINETT Sigma2 - the

National Infrastructure for High Performance Computing and Data Storage in Norway (NN9280K,

NS9001K, NS9599K). WRFb-CLM4.0 simulations were supported by computational time granted from

the National Infrastructures for Research and Technology S.A. (GRNET S.A.) in the National HPC





facility - ARIS - under project ID pr005025 and pr007033_thin. Edouard L. Davin and Ronny Meier
acknowledge financial support from the Swiss National Science Foundation (SNSF) through the
CLIMPULSE project and thank the Swiss National Supercomputing Centre (CSCS) for providing
computing resources. P. Hoffmann is funded by the Climate Service Center Germany (GERICS) of the
Helmholtz-Zentrum Hereon in the frame of the Helmholtz-Institut Climate Service Science (HICSS)
project LANDMATE. The authors gratefully acknowledge the WCRP CORDEX Flagship Pilot Study
LUCAS "Land use and Climate Across Scales" and the research data exchange infrastructure and
services provided by the Jülich Supercomputing Centre, Germany, as part of the Helmholtz Data
Federation initiative. R. M. Cardoso, D. C. A. Lima P. M. M. Soares were supported by national funds
through FCT (Fundação para a Ciência e a Tecnologia, Portugal) under project LEADING (PTDC/CTA-
MET/28914/2017), and project UIDB/50019/2020. This study contains modified Copernicus Climate
Change    Service    Information    2021.    ERA5-Land    data    are    available    at
https://doi.org/10.24381/cds.e2161bac  and  https://doi.org/10.24381/cds.68d2bb30.  The  information
related to GlobSnow data is presented in https://doi.org/10.1016/j.rse.2014.09.018. Variables from
MERRA2 have been downloaded in 2019 and 2020 via NASA/GSFC, Greenbelt, MD, USA, NASA
Goddard Earth Sciences Data and Information Services Center (GES DISC).

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



**Figures and Tables**

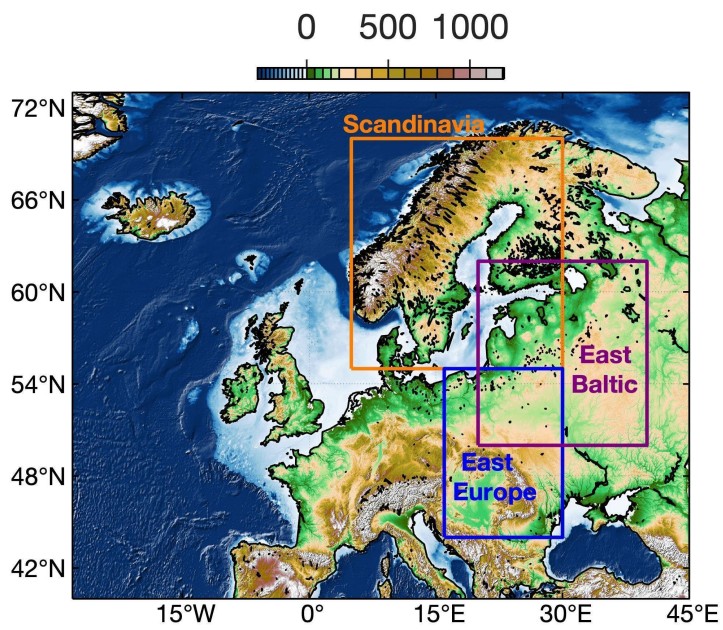


**Figure 1:** Map showing the location of the three regions of interest: Scandinavia (red), East Baltic (pink)
and East Europe (blue).



**Figure 2:** Spatial maps of SASI (Wm$^{-2}$) for satellite observations, the reanalysis ERA5-Land and the

ten regional climate simulations from the EVAL experiment of LUCAS from January to June, averaged

over the time period 1986-2015.

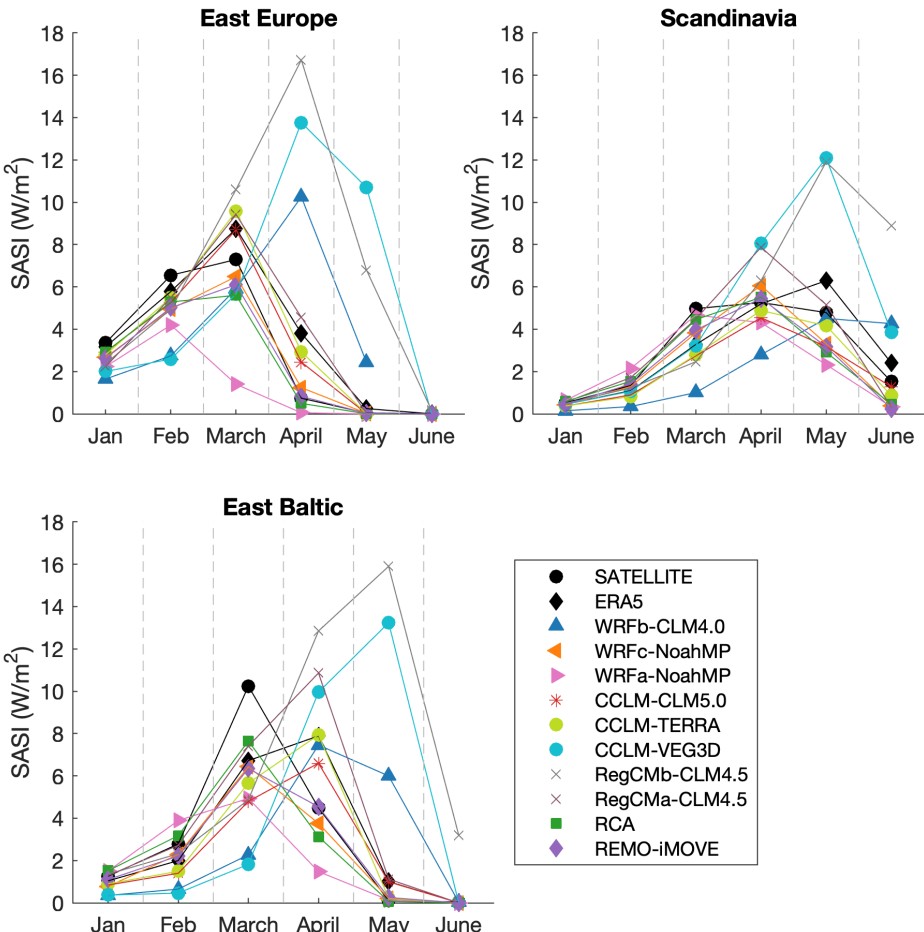

**Figure 3:** Time series of the spatial average of SASI for the satellite observations, the reanalysis ERA5-Land and the ten regional climate simulations from the EVAL experiment of LUCAS in Scandinavia, East Europe and East Baltic (see Figure 1 for their spatial extent). Data are averaged over the time period 1986-2015.


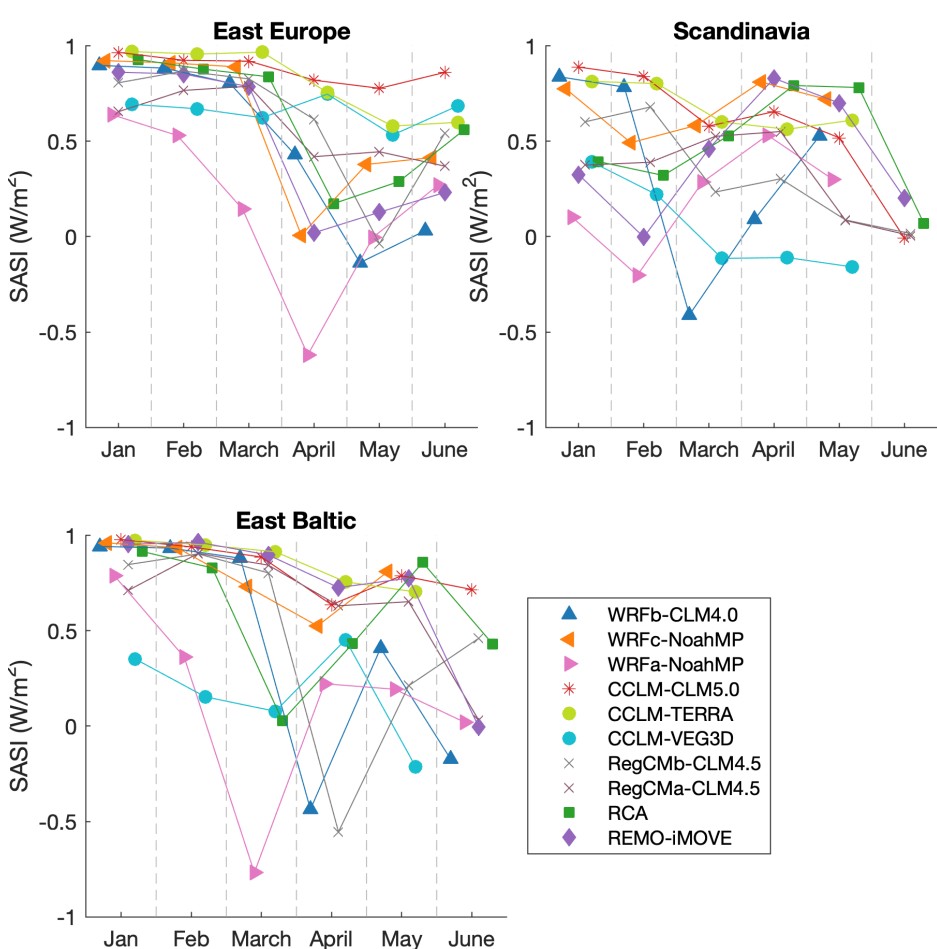

**Figure 4:** As in Figure 3 but for the pattern correlation between SASI and ERA5-Land for the LUCAS

simulations.





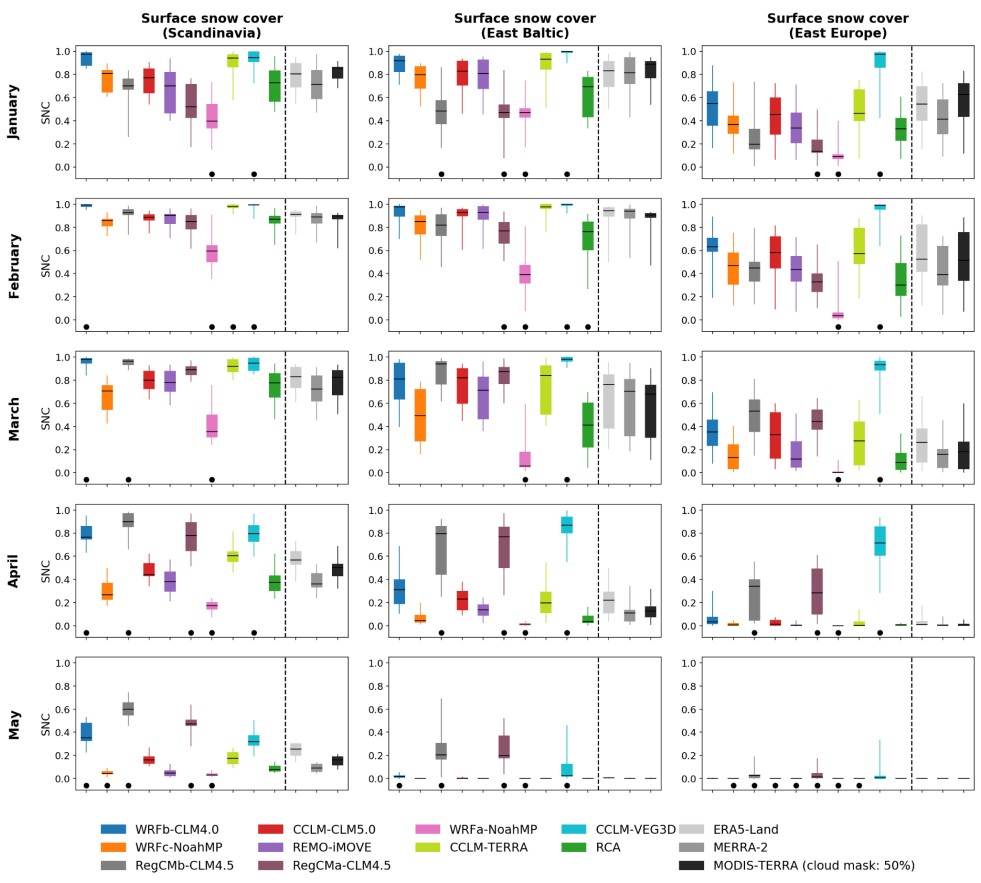

**Figure 5:** Snow cover for the 10 RCMs, MERRA-2, ERA5-Land, and MODIS satellite observations (using only data from days and pixels with less than 50% cloud cover) for January to May. The box-and-whisker-plots show the interannual variability of snow cover over 1986-2015, with the bar representing the median, boxes the interquartile range, and whiskers the minimum/maximum values. Dots indicate models lying outside the range of the reference datasets MERRA-2, ERA5-Land, and MODIS (i.e., the 25th (75th) model percentile is higher (lower) than the highest 75th (lowest 25th) quantile of the reference datasets).



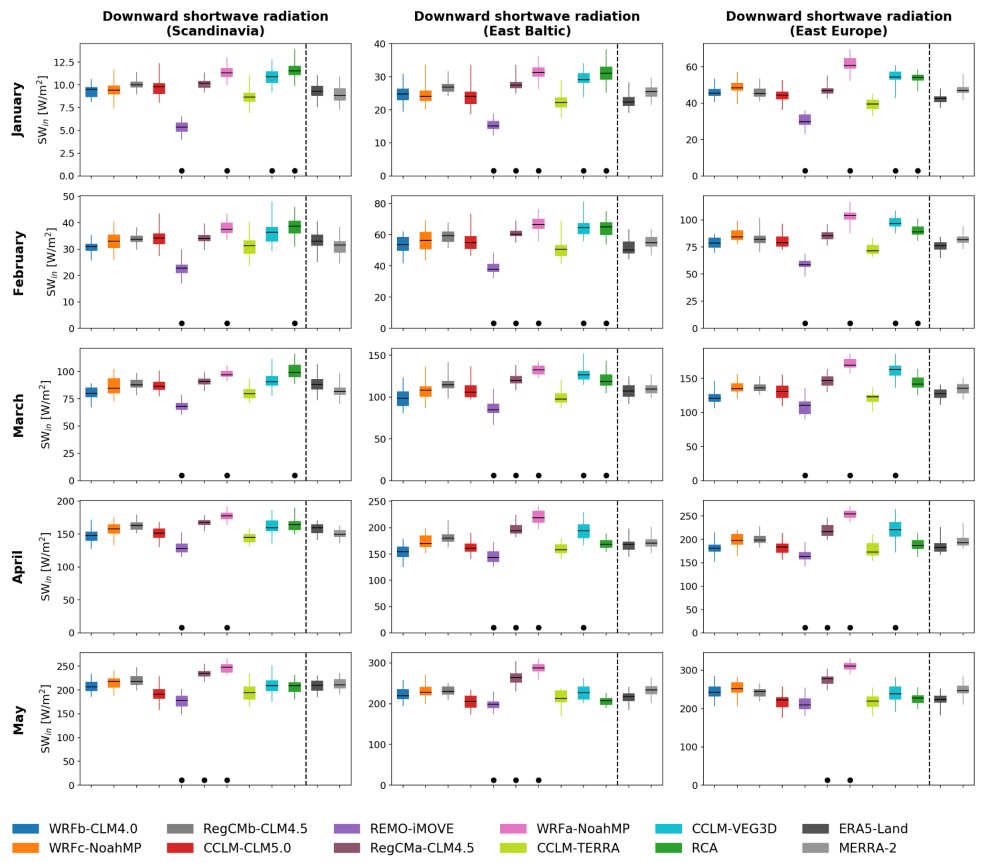

**Figure 6:** Downward surface shortwave radiation for the 10 RCMs for MERRA-2, and ERA5-Land, for January to May. The box-and-whisker-plots show the interannual variability of downward shortwave radiation over 1986-2015, with the bar representing the median, boxes the interquartile range, and whiskers the minimum/maximum values. Dots indicate models lying outside the range of the reference datasets MERRA-2, ERA5-Land, and MODIS (i.e., the 25th (75th) model percentile is higher (lower) than the highest 75th (lowest 25th) quantile of the reference datasets).



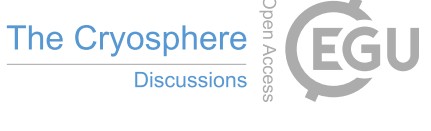

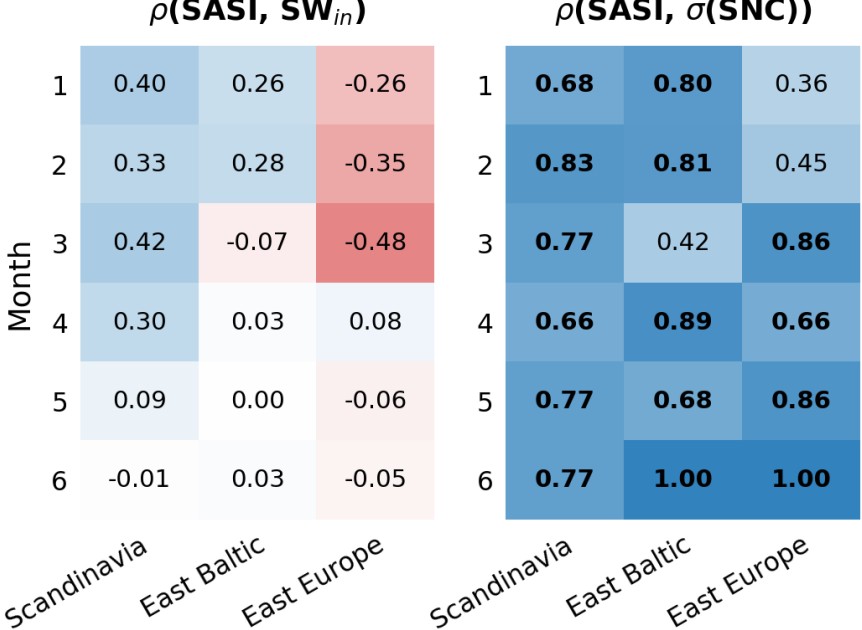

704

**Figure 7:** Pearson correlation between SASI and shortwave radiation (left), and SASI and standard

deviation of snow cover (right) calculated across RCMs for the three regions Scandinavia, East Baltic,

and East Europe for the months January to June during 1986-2015. The values represent the variable

(shortwave radiation or variability in snow cover) to which the inter-model variability of SASI is

predominantly related to. Bold values indicate statistical significance at the 0.05 level (two-tailed p-

value).










| Institute ID | RCM | LSM | Representation of sub-grid scale surface heterogeneity | Phenology | Snow- vegetation interaction | Name of the models |
|---|---|---|---|---|---|---|
| BCCR | WRF v3.8.1 [Skamarock et al., 2008] | NoahMP [Niu et al., 2011] | PFT-dominant | Prescribed | Deardorff, 1978; Niu and Yang, 2007 | WRFc-NoahMP |
| CUNI | RegCM v4.7 [Giorgi et al., 2012] | CLM4.5 [Oleson et al., 2013] | PFT-tile | Prescribed | Wang and Zeng, 2009 | RegCMb-CLM4.5 |
| ETH | Cosmo_5.0_clm9 [Soerland et al., 2021] | CLM5.0 [Lawrence et al., 2020] | PFT-tile | Prescribed | Wang and Zeng, 2009; Lawrence et al., 2020; van Kampenhout et al., 2017 | CCLM-CLM5.0 |
| GERICS | REMO2009 [Jacob et al., 2012] | iMOVE [Wilhelm et al., 2014] | PFT-tile | Interactive | Roeckner et al., 1996; Kotlarski, 2007 | REMO-iMOVE |
| ICTP | RegCM v4.6 [Giorgi et al., 2012] | CLM4.5 [Oleson et al., 2013] | PFT-tile | Prescribed | Wang and Zeng, 2009 | RegCMa-CLM4.5 |
| IDL | WRF v3.8.1D [Skamarock et al., 2008] | NoahMP [Niu et al., 2011] | PFT-dominant | Prescribed | Deardorff, 1978; Niu and Yang, 2007 | WRFa-NoahMP |
| KIT | Cosmo_5.0_clm9 [Soerland et al., 2021; Rockel et al., 2008] | VEG3D [Braun and Schädler, 2005] | PFT-dominant | Prescribed | Grabe, 2002 | CCLM-VEG3D |
| SMHI | RCA4 [Strandberg et al., 2015] | Internal [Samuelsson et al., 2006] | PFT-tile | Prescribed | Samuelsson et al., 2015 | RCA |
| AUTH | WRF v3.8.1 [Skamarock et al., 2008] | CLM4.0 [Oleson et al., 2010] | PFT-tile | Prescribed | Wang and Zeng, 2009 | WRFb-CLM4.0 |
| CLMcom-JLU | Cosmo_5.0_clm9 [Soerland et al., 2021] | TERRA-ML [Schrodin and Heise, 2002] | PFT-dominant | Prescribed | Doms et al., 2013 | CCLM-TERRA |

**Table 1:** Summary of participating RCMs and their LSMs.