# Peer review of "Land-atmosphere interactions in sub-polar and alpine climates in the CORDEX"

_The Cryosphere, 2021_

## Author Comment (AC1)

**Land-atmosphere interactions in sub-polar and alpine climates in the CORDEX FPS LUCAS models: I. Evaluation of the snow-albedo effect**

**Reviewer #1:**

**General comments**

The paper focuses on a snow-albedo sensitivity index (SASI), which describes interannual variations in surface net shortwave radiation resulting from anomalies in snow cover. The behavior of SASI is intercompared in a set of ten regional climate models (RCMs) from the LUCAS study, and it is also compared to satellite and reanalysis data.

It is shown that (1) SASI most typically peaks in the melting season; (2) there are substantial differences in the simulation of SASI among the models as well as between the models and observations; (3) the choice of the land-surface model can influence the intermodel differences in SASI substantially, but differences in other parameterizations such as convection or planetary boundary layer processes can also be important; (4) and the differences in SASI are more related to differences in (standard deviation of) snow cover than downwelling solar radiation in the models.

The coordinated LUCAS simulations represent a valuable dataset, and documenting the intermodel differences in snow conditions and the level of model-vs-observations

(dis)agreement is a worthy effort. I think there is potential for this paper to be published in The Cryosphere, but there are issues that should be carefully considered by the authors. In particular, I'm wondering if SASI is the most natural starting point for this paper. Would it not be better to start the story from the basics, that is the simulation of snow cover itself? Indeed, the motivation for considering SASI should be outlined more clearly. E.g., why is it important to compare the snow-related variability in the surface energy budget, when the systematic differences in snow cover between the models exceed the interannual variability?

Thank you for your interest in our work and your constructive comments. We agree with you that the organization of the article could be modified starting with the basics, as you suggested, looking at the simulation of the snow cover. This point was discussed several times by the authors ending in the decision to start with SASI, however, the organization you suggested makes more sense so we will modify it in the new version of the article, also clarifying the motivation for this work from the start.

In general, we believe that the modifications you suggested can be addressed. They will improve the quality of the article as some parts deserves more explanations, such as the Introduction better explaining the motivation of the article or the derivation of SASI with satellite observations. I will provide preliminary answers to the general and minor comments below. A more detailed answer to your comments will be provided with the new version of the manuscript at a later stage of the review process as we are currently working on your reviews.

**Major comments**

1. If/when this is the first snow-focused study on the LUCAS simulations, I think you should not start from a derived quantity (SASI) but more from the basics: document the snow cover

and perhaps also the snow water equivalent in the simulations. Plot(s) like Fig. 2 would do the job.

There are two reasons why dicussing the systematic snow cover differences would be important. The first point is their large effect on the surface energy budget and hence the simulated climate. For the sake of the argument, one could define a ``snow radiative forcing (SRF)'' or ``snow radiative effect (SRE)'' as a difference to the snow-free case:

SRF = - SW fsno $\Delta\alpha$

This is similar to the definition of SASI in Eq. (1) of the manuscript (and with the same notation), excecpt that the standard deviation of snow cover σ(fsno) is replaced by the mean value fsno for the given calendar month. Since the systematic intermodel differences in fsno are often substantially larger than the corresponding differences in σ(fsno) (which can be easily inferred from Fig. 5), it follows that the intermodel differences in SRF exceed those of SASI.

Second, showing the monthly climatology of snow cover in the simulations would help to explain much of the variations in SASI. Intuitively, interannual variations in snow cover for a given month/region are small in the cases in which the climatological snow cover fraction is close to either 1 or 0. The former applies e.g. to northern parts of Scandinavia in winter, and the latter to most regions in late spring and summer. Conversely, the interannual variations in snow cover (and hence also the values of SASI) are more likely to be large when the climatological snow cover fraction takes intermediate values. This applies to two cases. First, in the snowmelt period, snow cover fraction decreases rapidly. Therefore, interannual variations in snowmelt timing can result in large year-to-year variations in snow cover. Second, in the more southerly regions, snow cover in winter may be thin and intermittent (i.e., snow comes and goes). Consequently, due to variations in weather conditions, the interannual variations in snow cover can be large.

Yes, we agree with the reviewer, the organization of the article will be modified in the next version of the manuscript, starting with a comparison of the representation of snow cover. Furthermore, thank you very much for the detailed explanations provided here. We will do our best to address this point, better including this background information in the new version of the article.

2. In defining SASI, the assumption of a surface albedo difference of Δα=0.4 between snow-covered and snow-free land seems somewhat arbitrary. It is also not fully clear what is meant by snow cover fraction: does it include only the snow cover on land, or also snow on vegetation? Judging by section 2.1.3, the LSMs have different approaches, but it is not obvious from the text, what this means for fsno. Please try to clarify this.

Thank you, we agree that the different approaches of the LSMs should be clarified, we will modify the text including the information we can find on the subject. Furthermore, yes, we agree with the reviewer that the value of 0.4 as the average albedo difference between snow-covered and snow-free surfaces might not be ideal. The influence of the choice of this value can be tested and will be tested before we submit the manuscript again. Depending on the results we find we will keep or update this value and describe these sensitivity tests in the new version of the article.

I suggest that, to evaluate the robustness of your results, you compare the standard deviation of albedo assumed by the SASI formula (i.e., $0.4\sigma(fsno)$ ) with the actual standard deviation of monthly-mean albedo values $\sigma(\alpha)$. The monthly value of albedo could be calculated based on the values of downwelling and upwelling (or downwelling and net) SW radiation. Note that $\sigma(\alpha)$ may also be influenced by albedo variations due to other factors than snow (e.g. vegetation), but I would assume that in the winter/spring seasons, the interannual variations in surface albedo are overwhelmingly dominated by variations in snow conditions.

Thank you for this suggestion, we will do our best to realize this evaluation and include its results in the manuscript.

3. The explanations regarding the reasons for the intermodel differences remain rather vague. Perhaps it is not possible to go very deep with an ``ensemble of opportunity'' like the LUCAS simulations, where you have a very sparse matrix of RCM-LSM combinations. Nevertheless, I think the analysis could be clarified by considering more explicitly the three ``groups'' of models you have available (the WRF group with 3 models, the CCLM group with 3 models, and the RegCM group with 2 models). I would suggest one extra figure for each of the groups, showing the monthly (January-June) values of downward SW radiation, climatogical snow cover fsno, its standard deviation $\sigma(fsno)$ and SASI in different rows, and the three regions in different columns.

Thank you for this suggestion. It was indeed difficult to find a good way to talk about the intermodel differences as, as suggested by the reviewer, there is a very sparse mix of RCM-LSM combinations. However, the comparisons of WRF/CCLM/RegCM groups is feasible and could bring valuable information in the article.

Most of this information is already available in the figures, but not in a form in which the behavior of the models within each group can be compared easily. If you think this is too much for the main paper, placing these figures in the Supplementary material would be an option.

Thank you, the supplementary material seems like the best option in this case. We will add these figures in this section of the manuscript.

**Minor comments:**

1.       lines 34-35: I think that characterizing SASI as ``the radiative forcing due to the snow-albedo effect'' is misleading. At least to me, the most natural definition for the radiative forcing due to the snow-albedo effect would be the difference to the snow-free case (see major comment 1). If you want to call SASI a radiative forcing, then something like

``radiative forcing associated with interannual variations in the snow-albedo effect'' or ``radiative forcing associated with snow-cover anomalies'' is suggested.

Thank you for the suggestions, the description of SASI will be modified in the different part of the article mentioning it using one for your suggestions.

2.       lines 63, 66, 195, 652: The SASI index is not defined in Xu and Dirmeyer (2011), and neither in Xu and Dirmeyer (2013) (Journal of Hydrometeorology, pages 389–403). The correct reference would be Xu and Dirmeyer (2013) (Journal of Hydrometeorology, pages 404-418).

Thank you, this mistake has been corrected in the text.

3. lines 70 and 143: please add a reference for this statement (the impact of snow cover on precipitation is not obvious to me).

Thank you for pointing this out, references will be added for the sentences you suggested.

4. lines 75-76. Positive feedbacks amplify anomalies. Negative feedbacks act to damp them.

This sentence will be reformulated to clarify the effect of the feedbacks.

5. line 81. Radiative forcing associated with snow cover anomalies? See the first minor comment.

The description of SASI has now been reformulated following the suggestions from the minor comment #1.

6. lines 85-87. Other studies could also be mentioned. See, for example, Diro, G.T., Sushama, L. and Huziy, O. Snow-atmosphere coupling and its impact on temperature variability and extremes over North America. Clim Dyn 50, 2993-3007, https://doi.org/10.1007/s00382-017-3788-5, 2018.

Thank you, this reference has been added to the manuscript.

7. lines 115-116. It is not necessary mention the GRASS and FOREST experiments here (they are already mentioned on line 97-98).

Thank you, the part of the sentence mentioning GRASS and FOREST experiments has been removed.

8. lines 149--157: I find this description unclear. Given the definition of SASI (Eq. 1), the key questions here are how do the models define the snow cover fraction fsno and whether or not snow on vegetation is included in fsno.

This description will be modified in the new version of the text.

9. line 174: You also use the snow cover from ERA5-Land (in Fig. 5).

Thank you, this has been corrected in the text.

10. line 180: The use of ``two different thresholds (20% and 50%)'' immediately raises questions like why do you apply two thresholds, which of them do you apply in your figures, or is it perhaps case-dependent.

Thank you, this will be corrected in the next version of the text.

11. lines 190-191. To be sure, is this ``MODIS masking'' applied to all model results throughout the paper?

This point will be clarified in the next version of the text.

12. line 197: ``net radiation'' is wrong. It should be the downward radiation. But perhaps this is just a typo?

Thank you, this will be corrected in the next version of the text.

13.     line 197: I suppose standard deviation refers here to the interannual variation of monthly-mean values. Please be explicit about this.

This point has now been clarified in the text.

14.     lines 221-222: ``then decreasing when snow starts melting'' gives the impression that SASI reaches its maximum value right before the ablation period. But a comparison of SASI (in Figs. 2, 3), snow cover (Fig. 5) and SWE (Fig. S1) rather gives the impression that SASI peaks in the middle of the ablation period (which is what I would also assume based on physical reasoning).

Yes, thank you this will be modified in the next version of the article.

15.     lines 236-238, 247-249. Regarding the role of the atmospheric model, I am not sure if there is anything special about the convective or planetary boundary layer parameterizations as such; changes in other physical parameterizations such e.g. the cloud scheme could also be important. In general, I would expect that the impact from the atmospheric model comes mostly through the effects of precipitation and temperature. (the latter influencing both the phase of precipitation and snow melting). Have you looked at the differences in temperature and precipitation between WRFc-NoahMP and WRFa-NoahMP? Judging by Fig. 2 I would guess that WRFc-NoahMP either precipitates more, or features a colder climate in winter/spring than WRFa-NoahMP?

Yes, thank you this will be clarified in the next version of the article.

16.     line 241-242: "WRFa-NoahMP shows an earlier poleward migration of high SASI values compared to WRFb-CLM4.0". A plain language translation of this would be that snow melts earlier in WRFa-NoahMP!

Thank you, this sentence has been reformulated to clarify the this point.

17.     lines 267--268: ``The maximum in SASI marks the transition between the accumulation and ablation periods". In my understanding, the transition between the accumulation and ablation periods refers to the time when snow cover and SWE are at maximum. Your results suggest that SASI increases when snow starts to melt, and it is at maximum when snowmelt is well underway, i.e., definitely after the snow cover/SWE maximum. See also minor comment 14.

Yes, thank you this will be modified in the next version of the article.

18.     line 271: the later maximum of SASI for ERA5-Land than satellite data for East Baltic and Scandinavia is consistent with later snowmelt in ERA5-Land (as seen from Figs. 5 and S1). Incidentally, could that be related to the different data periods (1986-2015 vs. 2003-2015)?

Thank you, this will be clarified in the next version of the article.

19.     lines 273-276: A problem with this explanation is that East Baltic has lower elevations than East Europe.

Thank you for the clarification, this will be corrected in the next version of the article.

20.     lines 323-324: It is not clear what is meant with ``a common bias between the models''. Systematic differences between the models, or systematic differences between the models and observations?

Thank you, this point has been clarified in the text, reformulating the sentence.

21.     line 339: ``rate of snow melting'' or ``timing of snowmelt''? Also, specify explicitly that with melting, you refer here to the reduction of snow mass (SWE).

Thank you, this point has been clarified in the text, reformulating the sentence.

22.     line 368: Radiative forcing associated with interannual variations in snow cover?

Thank you, this point has been clarified in the text.

23.     line 370: replace ``albedo'' with ``surface net SW radiation''.

Thank you, albedo has been replaced with surface net SW radiation.

24.     line 382: Please specify what you mean with a ``common bias regarding snow cover''. Overestimation? Underestimation??

Yes, this point has been clarified in the text.

25.     line 387: How can you infer this from the available dataset, when there are presumably many other differences between the LSMs? What one could probably say is that there was no systematic difference between the PFT-dominant and PFT-tile models.

Yes, we agree with your comment. This sentence has been modified to include your suggestion.

26. The figures and table(s) should be organized in such a way that they support a visual comparison of simulations with the same model components (see major comment 3). Figure 2 is well-designed in this respect: the models/simulations within the WRF group, the CCLM group and the RegCM group can be easily compared. Please apply this ordering of simulations also in Figs. 5, 6 and S1 and in Table 1. In addition, Figures 3 and 4 could be improved by using, for simulations within each group, the same color but different symbols for the different simulations.

Thank you, these suggestions will be included in the new version of the figures.

27.     Fig. 2. As noted in the first major comment, I strongly recommend adding a similar figure for snow cover. Also, similar maps for the interannual standard deviation of snow cover fraction and the downwelling SW radiation would be useful for visually explaining the behavior of SASI. (If you think this increases the number of figures too much, the use of Supplementary material is always an option).

Thank you, these suggestions will be included in the new version of the manuscripts as we are currently working on it.

28.     Fig. 4. The y-axis labels are wrong (it is correlation, which is unitless. Also, I'm not fully convinced this figure is necessary in the first place.

Thank for you for noticing this, we will modify the y-axis and see if this figure is necessary in the new version of the article.

**Technical and language corrections**

1.      line 107: ``Section 4 the last sections"

Thank you this mistake has been corrected in the text.

2.      line 111: Delete the latter ``simulations".

Thank you, the latter simulation has been removed.

3.      line 159: Replace ``counts" with ``includes"?

Thank you, correction made in the text.

4.      line 165: Replace ``first very" with ``very first".

Thank you, this is corrected now.

5.      lines 318-322: This could be streamlined. ``In January, WRFa-NoahMP simulates consistently the least snow cover in the three regions (0.4 for Scandinavia, 0.3 for East Baltic, and 0.1 for East Europe), while WRFa-CLM4.0 simulates the largest snow cover (1.0 in all three regions)."

Thank you, this sentence has been modified in the text.

6.      Fig. 2. Can anything be done to the strange land mask in CCLM-TERRA?

We will do our best to address this point in the next version of the article.

---

## Author Comment (AC3)

**Land-atmosphere interactions in sub-polar and alpine climates in the CORDEX FPS LUCAS models: I. Evaluation of the snow-albedo effect**

**Reviewer #2:**

**General comments:**

In this paper snow albedo effect is studied in the Europe in winter and spring time. Simulations from Regional Climate Models are used to produce Snow Albedo Sensitivity Index (SASI) and these RCM based SASI values are compared to the SASI values derived from reanalysis and satellite observations. Conclusions are that accurate retrieval of SASI is more dependent of correct snow cover simulations than chosen atmospheric models. This leads to the observation that choosing correct Land Surface Model have an important role in simulating snow albedo effect. The subject itself is very interesting, and this study would be a good fit to The Cryosphere journal. However, I have some concerns.

*Thank you very much for your interest in our work and your constructive comments. We believe that the modifications you suggested can be addressed. They will improve the quality of the article as some parts are unclear such as the derivation of SASI with satellite observations or the description of the LUCAS experiments. I will provide preliminary answers to the specific and minor comments below. A more detailed answer to your comments will be provided with the new version of the manuscript at a later stage.*

**Specific comments:**

- I have some difficulties to understand how the satellite based SASI is formed. First, where the SW data is from? Secondly, if snow cover extent from MODIS is used, and that covers years 2003-2015, then I assume that satellite based SASI is covering years 2003-2015. Therefore, satellite based SASI cannot be used to verify/compare RCMs based SASI covering year 1986-2015. Due to the climate change, snow cover extent is vastly different in 2000s than in 1980s or 1990s (e.g. https://www.ncdc.noaa.gov/snow-and-ice/extent/snow-cover/nhland/4). I have doubts that weighting every grid point by the amount of MODIS data is enough to make satellite based SASI comparable to reanalysis and model based SASI.

*Yes, we agree that the section describing the calculation of SASI based on satellite observations deserves more explanations as this is a complex calculation. This will be provided in the next version of the article. We will also further discuss the comparability of SASI based on satellite observations with SASI based on reanalysis and regional climate model outputs. We agree that this is also a point that deserves more explanations.*

- The value 0.4 as the average albedo difference between a snow-covered and snow-free surfaces is problematic. First, chosen three areas have different vegetation. The Scandinavian area are mostly boreal forest (needleleaved evergreen forest) whereas the East Baltic and the East Europe have more deciduous trees. Throughout the year needleleaved evergreen trees have their "leaves" on, but deciduous trees don't. So difference between snow-covered and snow-free surface albedo should be different in the Scandinavian area (high-latitudes) compared to the two other regions (midlatitudes). Also, the difference depends on whether snow is new or old. Old snow can

have impurities, which lowers albedo (Warren and Wiscombe, 1980)). And, in winter snow can sporadically accumulate on trees, which itself increases albedo. I suggest that authors modify â□□α corresponding better different scenarios.

Yes, we agree with the reviewer that the value of 0.4 as the average albedo difference between snow-covered and snow-free surfaces might not be ideal. The influence of the choice of this value can be tested and will be tested before we submit the manuscript again. Depending on the results we find we will keep or update this value and describe these sensitivity tests in the new version of the article.

**Minor comments:**

The description of LUCAS experiments will need some clarifications and more details. It is of course allowed to specify manuscript to people with certain scientific knowledge, but as not every reader is familiar with climate models, it would be reader-friendly to provide more explanations. What is a rotated coordinate system, could that term be explained? What is the time resolution of the simulations? Hourly, daily, monthly? In line 121 is said that "outputs from ten … RCM simulations", are there more than
those chosen ten? If yes, why those specific ten simulations are chosen?
Table 1, could you open the used acronyms in Table 1 caption?
Are RCMs WRF 3.8.1 and WRF 3.8.1D the same? If not, what is the difference?
Thank you for these questions/comments, it was difficult to determine the level of information needed for the description of the LUCAS experiments. These points will be addressed, more information will be included in the next version of the article.

Lines 158-168: snow schemes of CLM versions, Noah-MP and RCA4 system are described, but what about iMOVE, VEG3D and TERRA-ML?
More information will be provided in the next version of the article.

Lines 180-181: the two thresholds for cloud cover are used. 50% of cloud cover is quite a lot of clouds in one cell, why this threshold was chosen? How were these thresholds used?
We have tested several thresholds and will provide more information on this point in the next version of the article.

Line 182: why also "good" and "ok" flagged data was used?
Yes, this point deserves more explanation and will be clarified in the next version of the article.

Line 209: mention that the Scandinavian region have mostly needleleaved evergreen forest, whereas other two regions have more deciduous trees.
Yes, thank you, this point will be included in the next version of the article.

line 277: The peaks are quite pronounced in the East Europe and the East Baltic regions, but I think they are less pronounced in the Scandinavian region. Could it be due to the illumination conditions?
The illumination might be part of the explanation yes, we will comment this point in the next version of the article.
line 312: based on Figure 5, the snow cover for MODIS is from MODIS-TERRA, is that correct? Why MODIS-TERRA, if you also have MODIS-AQUA data?
Yes, this needs to be clarified in the next version of the article.
line 314: what are those limitations and biases that are referenced to in this sentence?
The reasoning behind this sentence will be expanded in the next version of the article.

line 328: also WRFb-CLM4.0 have high values during the ablation period. Should that model also be added?

Yes, this point should be mentioned in the article. Thank you.

lines 322-334: would it be more informative to add different markers whether models are over or under the range of reference datasets? For example, black dots when over and red?) x when under?

Thank you for the suggestion, we can test it and see if this option is more informative or if it becomes too messy.

line 346- 349: I would argue that REMO-iMOVE and WRFa-NoahMP have very different results, not REMO-iMOVE and CCLM-VEG3D, if these results are based on Figure 5. But also, based on Figure 3, CCLM-VEG3D do not reproduce SASI well at all.

We will come back to this part of the text and provide clarifications.

Figure2: colorbar ticks and color limits do not match, could it be modified?

Yes, this will be modified in the next version of the article.

Figure 3 and 4: can horizontal lines be added? It would make reading of the figures much easier. Also, it would be more informative to draw ERA5 and SATELLITE lines last so they would be top of everything.

Thank you, this is a good point, we will work on these modifications.

Figure 5: black color of MODIS-TERRA, especially black median line in the very dark grey bar is difficult to see. Can bar be made more lighter grey?

Yes, this can be modified in the next version of the article.

Table 1: Can table rows be listed based on RCM (as in Figure 2), not institute? It would be easier to read.

Yes, this can be addressed in the next version of the article.

**Technical corrections:**

lines 46-47: word "it" is ambiguous, could this sentence be modified to be easier to read?

Yes, this sentence will be modified.

line 80: open the "RCMs" acronym

Thank you, this will be corrected in the new version of the article.

line 107: "Section4 the last sections" -> remove "the last section"

Thank you, this will be corrected in the new version of the article.

line 175: add MODIS product names (for TERRA: MOD10C1 and for AQUA: MYD10C1)

Thank you, they will be included in the new version of the article.

line 176: reference to the same section 2.2 is not necessary, remove it or change it

This reference will be removed.

line 225: "The model data.." -> change it to "Most of the model data …"

This sentence will be modified.

line 319: ".. snow cover varies between…" -> "..snow cover mean varies between…"

This sentence will be corrected.

**References:**

Warren, S. G., & Wiscombe, W. J. (1980). A Model for the Spectral Albedo of Snow. II: Snow Containing Atmospheric Aerosols, Journal of Atmospheric Sciences, 37(12), 2734-2745. Retrieved Nov 15, 2021, from https://journals.ametsoc.org/view/journals/atsc/37/12/1520-0469_1980_037_2734_amftsa_2_0_co_2.xml

---

## Author Response (AR1)

Cryosphere: Manuscript tc-2021-290:

**Land-atmosphere interactions in sub-polar and alpine climates in the CORDEX FPS LUCAS models: I. Evaluation of the snow-albedo effect**

**Reviewer #1:**

**General comments**

The paper focuses on a snow-albedo sensitivity index (SASI), which describes interannual variations in surface net shortwave radiation resulting from anomalies in snow cover. The behavior of SASI is intercompared in a set of ten regional climate models (RCMs) from the LUCAS study, and it is also compared to satellite and reanalysis data.

It is shown that (1) SASI most typically peaks in the melting season; (2) there are substantial differences in the simulation of SASI among the models as well as between the models and observations; (3) the choice of the land-surface model can influence the intermodel differences in SASI substantially, but differences in other parameterizations such as convection or planetary boundary layer processes can also be important; (4) and the differences in SASI are more related to differences in (standard deviation of) snow cover than downwelling solar radiation in the models.

The coordinated LUCAS simulations represent a valuable dataset, and documenting the intermodel differences in snow conditions and the level of model-vs-observations (dis)agreement is a worthy effort. I think there is potential for this paper to be published in The Cryosphere, but there are issues that should be carefully considered by the authors. In particular, I'm wondering if SASI is the most natural starting point for this paper. Would it not be better to start the story from the basics, that is the simulation of snow cover itself? Indeed, the motivation for considering SASI should be outlined more clearly. E.g., why is it important to compare the snow-related variability in the surface energy budget, when the systematic differences in snow cover between the models exceed the interannual variability?

Thank you very much for your suggestions and comments on our work, which improved the quality of the article. We have worked a lot on the article, starting by the Introduction where we try to formulate better the motivation for this work. We also modified the methodology section to better present the models and their different specificities. Then, as suggested by the reviewer, we start the result section with a comparison of the representation of snow cover for the satellite observations, reanalyses and LUCAS simulations. This is indeed a more natural way to start the article, leading to SASI and the importance of the representation of snow-atmosphere processes in climate models. More details on this specific point are coming in Major comment #1.

**Major comments**

1. If/when this is the first snow-focused study on the LUCAS simulations, I think you should not start from a derived quantity (SASI) but more from the basics: document the snow cover and perhaps also the snow water equivalent in the simulations. Plot(s) like Fig. 2 would do the job.

There are two reasons why discussing the systematic snow cover differences would be important. The first point is their large effect on the surface energy budget and hence the simulated climate. For the sake of the argument, one could define a ``snow radiative forcing (SRF)'' or ``snow radiative effect (SRE)'' as a difference to the snow-free case:

SRF = - SW fsno Δα

This is similar to the definition of SASI in Eq. (1) of the manuscript (and with the same notation), excecpt that the standard deviation of snow cover σ(fsno) is replaced by the mean value fsno for the given calendar month. Since the systematic intermodel differences in fsno are often substantially larger than the corresponding differences in σ(fsno) (which can be easily inferred from Fig. 5), it follows that the intermodel differences in SRF exceed those of SASI.

Second, showing the monthly climatology of snow cover in the simulations would help to explain much of the variations in SASI. Intuitively, interannual variations in snow cover for a given month/region are small in the cases in which the climatological snow cover fraction is close to either 1 or 0. The former applies e.g. to northern parts of Scandinavia in winter, and the latter to most regions in late spring and summer. Conversely, the interannual variations in snow cover (and hence also the values of SASI) are more likely to be large when the climatological snow cover fraction takes intermediate values. This applies to two cases. First, in the snowmelt period, snow cover fraction decreases rapidly. Therefore, interannual variations in snowmelt timing can result in large year-to-year variations in snow cover. Second, in the more southerly regions, snow cover in winter may be thin and intermittent (i.e., snow comes and goes). Consequently, due to variations in weather conditions, the interannual variations in snow cover can be large.

Thank you for the suggestion, the manuscript now starts the result section with a new part called "3.1 Snow cover in satellite observations, reanalysis and RCMs over Europe". This new section provides a comparison of the representation of snow cover in climate models with satellite observations and reanalyses. To realize this comparison, we included maps of snow cover for all datasets, in a new Figure 2 and associated Figure S2 (with all datasets averaged over 2003-2015). As you suggested, this section aims at discussing the representation of snow cover in the different datasets. We talk about the systematic differences in snow cover in models versus observations and their large effect on the surface energy budget and hence the simulated climate. We also refer to a new figure added to the supplemental material (S3) showing snow depth for the models and reanalysis. This section is a nice introduction leading to the examination of the ability of climate models to represent snow-atmosphere interactions, via the analysis of SASI in the different datasets (Section 3.2).

2. In defining SASI, the assumption of a surface albedo difference of Δα=0.4 between snow-covered and snow-free land seems somewhat arbitrary.

Looking at Figure 4 from Atlaskina et al. (2015), showing the mean albedo over snow-covered land surfaces in the Northern Hemisphere as presented in Figure R1, the assumption of 0.4 for the difference between snow-covered and snow-free areas seems realistic.

[Figure]

*Figure R1: Time series of mean domain albedo (Northern Hemisphere) for the years 2000-2013.*

Atlaskina, K., Berninger, F., and de Leeuw, G.: Satellite observations of changes in snow-covered land surface albedo during spring in the Northern Hemisphere, The Cryosphere, 9, 1879–1893, https://doi.org/10.5194/tc-9-1879-2015, 2015.

It is also not fully clear what is meant by snow cover fraction: does it include only the snow cover on land, or also snow on vegetation? Judging by section 2.1.3, the LSMs have different approaches, but it is not obvious from the text, what this means for fsno. Please try to clarify this.

 Section 2.1.3 has been modified to make it clearer, focus on snow schemes and snow cover fraction calculation.

 "All LSMs in the LUCAS ensemble derive the fraction of vegetation buried by snow, adopting similar approaches that account for snow depth, vegetation height and snow cover fraction. The snow cover fraction $f_{sno}$ measures the snow amount in water equivalent accumulated at the surface over bare soil or vegetation and influences calculation of surface albedo and fluxes. Canopy-intercepted snow does not contribute to the snow cover fraction at the ground. The CLM models (CLM4.0, CLM4.5 and CLM5.0; Swenson and Lawrence, 2012) and the internal LSM in the RCA4 model (Samuelsson et al., 2015) separately calculate the snow cover fraction during snowfall and snow melting processes, accounting for sub-grid orography when snow melting occurs. In NoahMP, the snow cover fraction depends on snow depth, ground roughness length and snow density (Niu and Yang, 2004). In VEG3D, the snow cover fraction is internally calculated as a function of snow depth and vegetation height and used to update surface parameters such as albedo. However, since fsno is not a default model output, the snow cover fraction has been computed for analysis purpose as a snow flag in case of a snow height above a certain threshold, producing a value that is equal to one or zero (i.e., the grid box is covered by snow or not).

 In the ensemble, some LSMs contain more sophistication than others. CLM5.0 (Lawrence et al., 2020) and NoahMP (Niu et al., 2007) treats separately canopy-intercepted snow and more realistically captures temperature and wind effects on snow processes. In addition, LSMs differ in the number of additional layers for snow calculation: CLM5.0 uses 12 snow layers; CLM4.0, CLM4.5 and TERRA-ML (Tolle et al., 2018) use five, three in NoahMP, and two in VEG3D and iMOVE, and one in the RCA4 model. The iMOVE model adopts the snow parameterisation from the global climate model ECHAM4 (Roeckner, et al., 1996) and reproduces the snow

albedo as a linear function of the snow surface temperature and of the forest fraction in a grid cell, with fixed maximum and minimum snow albedo at temperatures lower than -10°C and at 0°C, respectively (Kotlarskis, 2007). In the VEG3D model, the snow scheme is based on the Canadian Land Surface Scheme (CLASS) (Verseghy, 1991) and ISBA (Douville et al., 1995) and accounts for changes of surface albedo and emissivity as well as processes like compaction, destructive metamorphosis, the melting of snow, and the freezing of liquid water."

Douville, H., J-F. Royer, and J-F. Mahouf. 1995. A new snow parameterization for the Meteo-France climate model. Part I: Validation in stand-alone experiments. *Climate Dyn.* 12:21–35.

Roeckner, E., Arpe, K., Bentsson, L., Christoph, M., Claussen, M., Dümenil, L., Esch, M., Giorgetta, M., Schlese, U., and Schulzweida, U.: The atmospheric general circulation model ECHAM-4: Model description and simulation of present day climate. *Max-Planck Institut für Meteorologie Report* No. 218, 90 pp, 1996.

Tölle, M. H., M. Breil, K. Radtke, and H. J. Panitz, 2018: Sensitivity of European temperature to albedo parameterization in the regional climate model COSMO-CLM linked to extreme land use changes. Front. Environ. Sci., 6, 123, 10.3389/fenvs.2018.00123

I suggest that, to evaluate the robustness of your results, you compare the standard deviation of albedo assumed by the SASI formula (i.e., $0.4\sigma$(fsno) ) with the actual standard deviation of monthly-mean albedo values $\sigma(\alpha)$. The monthly value of albedo could be calculated based on the values of downwelling and upwelling (or downwelling and net) SW radiation. Note that $\sigma(\alpha)$ may also be influenced by albedo variations due to other factors than snow (e.g. vegetation), but I would assume that in the winter/spring seasons, the interannual variations in surface albedo are overwhelmingly dominated by variations in snow conditions.

Figure R2 presents the standard deviation of monthly albedo values for all the LUCAS simulations over Europe. Even if albedo variations can be due to other factors than snow, we would also expect these variations to be dominated by the changes in snow conditions, and therefore have a larger effect on SASI, confirming our findings. Here, we can take the example of CCLM-VEG3D, which presents high values in standard deviation of monthly albedo values, this is coherent with what we see in Figure 4, where this model presents high values of SASI.

[Figure]

Figure R2: Standard deviation of monthly mean albedo values for all LUCAS simulations.

3. The explanations regarding the reasons for the intermodel differences remain rather vague. Perhaps it is not possible to go very deep with an ``ensemble of opportunity'' like the LUCAS simulations, where you have a very sparse matrix of RCM-LSM combinations. Nevertheless, I think the analysis could be clarified by considering more explicitly the three ``groups'' of models you have available (the WRF group with 3 models, the CCLM group with 3 models,

and the RegCM group with 2 models). I would suggest one extra figure for each of the groups, showing the monthly (January-June) values of downward SW radiation, climatogical snow cover fsno, its standard deviation σ(fsno) and SASI in different rows, and the three regions in different columns.

Most of this information is already available in the figures, but not in a form in which the behavior of the models within each group can be compared easily. If you think this is too much for the main paper, placing these figures in the Supplementary material would be an option.

Thank you for these suggestions, all the figures of the article as well as Table 1 have been reorganized, they are now classified by "modeling groups". Thanks to the new organization of the figures, the text is trying to provide more comparison of the results between the different modeling groups, hopefully providing more insights on the inter model differences. Some additional figures have also been added to the supplemental material. However, not all the suggested figures were included as we have now a high number of figures in the manuscript.

**Minor comments:**

1.      lines 34-35: I think that characterizing SASI as ``the radiative forcing due to the snow-albedo effect" is misleading. At least to me, the most natural definition for the radiative forcing due to the snow-albedo effect would be the difference to the snow-free case (see major comment 1). If you want to call SASI a radiative forcing, then something like

``radiative forcing associated with interannual variations in the snow-albedo effect" or ``radiative forcing associated with snow-cover anomalies" is suggested.

Thank you for the suggestions, the description of SASI has been modified in the different parts of the article mentioning it, using the following definition: "SASI quantifies the radiative forcing associated with snow cover anomalies."

2.      lines 63, 66, 195, 652: The SASI index is not defined in Xu and Dirmeyer (2011), and neither in Xu and Dirmeyer (2013) (Journal of Hydrometeorology, pages 389–403). The correct reference would be Xu and Dirmeyer (2013) (Journal of Hydrometeorology, pages 404-418).

Thank you, this error has been corrected in the text.

3.      lines 70 and 143: please add a reference for this statement (the impact of snow cover on precipitation is not obvious to me).

Thank you for pointing this out, the following reference has been added for the sentences you suggested: Snyder et al. (2004): *Evaluating the influence of different vegetation biomes on the global climate.*

4.      lines 75-76. Positive feedbacks amplify anomalies. Negative feedbacks act to damp them.

This sentence has been reformulated to clarify the effect of the feedbacks.

5.      line 81. Radiative forcing associated with snow cover anomalies? See the first minor comment.

The description of SASI has now been reformulated based on minor comment #1.

6.      lines 85-87. Other studies could also be mentioned. See, for example, Diro, G.T., Sushama, L. and Huziy, O. Snow-atmosphere coupling and its impact on temperature variability and extremes over North America. Clim Dyn 50, 2993-3007, https://doi.org/10.1007/s00382-017-3788-5, 2018.

Thank you, this reference has been added to the manuscript.

7.      lines 115-116. It is not necessary to mention the GRASS and FOREST experiments here (they are already mentioned on line 97-98).

Thank you, the part of the sentence mentioning GRASS and FOREST experiments has been removed.

8.      lines 149--157: I find this description unclear. Given the definition of SASI (Eq. 1), the key questions here are how do the models define the snow cover fraction fsno and whether or not snow on vegetation is included in fsno.

Section 2.1.3 has been simplified to make it clearer, focus on snow schemes and snow cover fraction calculation. For more details, see major comment #2.

9.      line 174: You also use the snow cover from ERA5-Land (in Fig. 5).

Thank you, this error has been corrected in the text.

10.     line 180: The use of ``two different thresholds (20% and 50%)'' immediately raises questions like why do you apply two thresholds, which of them do you apply in your figures, or is it perhaps case-dependent.

In the main part of the paper we only show results using a threshold of 50%. This is now clarified in the text. To examine how sensitive the results are to the choice of thresholds, we also added the snow cover fraction obtained by selecting a threshold of 20% to the MODIS data in Supplementary Figure S1. The results are very similar but with a slight tendency of higher snow cover fractions when using a threshold of 20%.

11.     lines 190-191. To be sure, is this ``MODIS masking'' applied to all model results throughout the paper?

We have tested the impact of the specific masking that we applied to MODIS data against using monthly average MODIS data and found that the difference is small compared to the interannual variability of MODIS snow cover data. We thus decided to use monthly MODIS averages without applying a specific masking. We show results from both MODIS-TERRA and MODIS-AQUA in the supplemental material (Figure S1). The text in the methods has been adjusted accordingly:

"Since heavy cloud cover prevents a correct estimation of snow cover, data are masked according by applying a threshold of 50 % to the percent of clouds in each cell. For comparison, we show the results when applying a threshold of 20% in Supplementary Figure S1."

12.     line 197: ``net radiation'' is wrong. It should be the downward radiation. But perhaps this is just a typo?

Thank you, this was indeed a typo, which has been corrected in the text.

13. line 197: I suppose standard deviation refers here to the interannual variation of monthly-mean values. Please be explicit about this.

Yes, thank you. This point has been clarified in the text.

14. lines 221-222: ``then decreasing when snow starts melting'' gives the impression that SASI reaches its maximum value right before the ablation period. But a comparison of SASI (in Figs. 2, 3), snow cover (Fig. 5) and SWE (Fig. S1) rather gives the impression that SASI peaks in the middle of the ablation period (which is what I would also assume based on physical reasoning).

Thank you for this comment, you are right. This point will be clarified in the text, in the different places where we mention the peak in SASI.

15. lines 236-238, 247-249. Regarding the role of the atmospheric model, I am not sure if there is anything special about the convective or planetary boundary layer parameterizations as such; changes in other physical parameterizations such e.g. the cloud scheme could also be important. In general, I would expect that the impact from the atmospheric model comes mostly through the effects of precipitation and temperature. (the latter influencing both the phase of precipitation and snow melting). Have you looked at the differences in temperature and precipitation between WRFc-NoahMP and WRFa-NoahMP? Judging by Fig. 2 I would guess that WRFc-NoahMP either precipitates more, or features a colder climate in winter/spring than WRFa-NoahMP?

Yes, we agree with your comment that other parameterizations such as the cloud scheme could impact the representation of temperature, precipitation and therefore snow variables. However, for this specific case, the main difference between these two WRF configurations is the convective and planetary boundary layer parameterizations, that's why we think it can explain some of the differences we see. We did not look at the representation of precipitation in the models (and we don't currently have this variable). Unfortunately, previous articles based on the LUCAS simulations focused on the GRASS and FOREST simulations so this information is not included in previous publications. However, we will modify the text, better explaining the specific case of these two configurations and the potential role of these two parameterizations. Furthermore, we agree with you that the difference in parameterizations can affect the representation of temperature and precipitation, and that there could be an effect on the representation of snow, which would in return impact the representation of SASI. This has also ben clarified in the text.

16. line 241-242: "WRFa-NoahMP shows an earlier poleward migration of high SASI values compared to WRFb-CLM4.0". A plain language translation of this would be that snow melts earlier in WRFa-NoahMP!

Thank you, this sentence has been reformulated and should be clearer now.

17. lines 267--268: ``The maximum in SASI marks the transition between the accumulation and ablation periods''. In my understanding, the transition between the accumulation and ablation periods refers to the time when snow cover and SWE are at maximum. Your results suggest that SASI increases when snow starts to melt, and it is at maximum when snowmelt is well underway, i.e., definitely after the snow cover/SWE maximum. See also minor comment 14.

Thank you, similarly to minor comment 14, we will modify the text to clarify this point.

18.     line 271: the later maximum of SASI for ERA5-Land than satellite data for East Baltic and Scandinavia is consistent with later snowmelt in ERA5-Land (as seen from Figs. 5 and S1). Incidentally, could that be related to the different data periods (1986-2015 vs. 2003-2015)?

We tested this hypothesis by looking at snow cover on the same time period (2003-2015) for all datasets, as shown in the Supplementary Material, Figure S1. The different time period does not explain the later maximum in SASI for ERA5-Land compared to satellite observations.

19.     lines 273-276: A problem with this explanation is that East Baltic has lower elevations than East Europe.

Thank you for pointing out this mistake, the related sentence has been deleted.

20.     lines 323-324: It is not clear what is meant with ``a common bias between the models''. Systematic differences between the models, or systematic differences between the models and observations?

Here we mean the systematic differences between the models and observations, this has been clarified in the text.

21.     line 339: ``rate of snow melting'' or ``timing of snowmelt''? Also, specify explicitly that with melting, you refer here to the reduction of snow mass (SWE).

Thank you, this point has been clarified in the text, reformulating the related sentence.

22.     line 368: Radiative forcing associated with interannual variations in snow cover?

Thank you, this sentence has been modified based on your suggestion.

23.     line 370: replace ``albedo'' with ``surface net SW radiation''.

Thank you, albedo has been replaced with surface net SW radiation.

24.     line 382: Please specify what you mean with a ``common bias regarding snow cover''. Overestimation? Underestimation??

Thank you, this sentence deserves clarification. A common bias is not the right term as the different models do not show the same bias (overestimation/underestimation) but they almost all show a bias. Therefore, we reformulated the sentence using "systematic bias".

25.     line 387: How can you infer this from the available dataset, when there are presumably many other differences between the LSMs? What one could probably say is that there was no systematic difference between the PFT-dominant and PFT-tile models.

Yes, this sentence has been reformulated based on your suggestion.

26. The figures and table(s) should be organized in such a way that they support a visual comparison of simulations with the same model components (see major comment 3). Figure 2 is well-designed in this respect: the models/simulations within the WRF group, the CCLM group and the RegCM group can be easily compared. Please apply this ordering of simulations also in Figs. 5, 6 and S1 and in Table 1. In addition, Figures 3 and 4 could be improved by

using, for simulations within each group, the same color but different symbols for the different simulations.

Thank you for the suggestion, Table 1 as well as the figures have now been organized in the way you suggested, with groups of models. We agree that this is easier to read and hopefully will help the readers follow the analysis more easily.

27.     Fig. 2. As noted in the first major comment, I strongly recommend adding a similar figure for snow cover. Also, similar maps for the interannual standard deviation of snow cover fraction and the downwelling SW radiation would be useful for visually explaining the behavior of SASI. (If you think this increases the number of figures too much, the use of Supplementary material is always an option).

Thank you for the suggestion. We have added a new section (3.1) where we present the spatial maps of snow cover, similarly to what we have done for SASI. We also added other figures in the supplemental material such as interannual variations of snow depth (Figure S3) or snow cover (Figure S2) in the supplemental material. For downwelling SW radiation, we already had a figure in the article, see Figure 6.

28.     Fig. 4. The y-axis labels are wrong (it is correlation, which is unitless. Also, I'm not fully convinced this figure is necessary in the first place.

This figure has been removed, we agree that the text does not need a figure to support the explanation.

**Technical and language corrections**

1.     line 107: ``Section 4 the last sections"

Thank you, this mistake has been corrected in the text.

2.     line 111: Delete the latter ``simulations".

Thank you, the latter simulation has been removed.

3.     line 159: Replace ``counts" with ``includes"?

Thank you, correction made in the text.

4.     line 165: Replace ``first very" with ``very first".

Thank you, this is corrected now.

5.     lines 318-322: This could be streamlined. ``In January, WRFa-NoahMP simulates consistently the least snow cover in the three regions (0.4 for Scandinavia, 0.3 for East Baltic, and 0.1 for East Europe), while WRFa-CLM4.0 simulates the largest snow cover (1.0 in all three regions)."

Thank you for the suggestion, the modification has been modified in the text.

6.     Fig. 2. Can anything be done to the strange land mask in CCLM-TERRA?

CCLM-TERRA only provides snow cover data for grid cells that are considered as land by the model. Thus, the land mask of CCLM-TERRA is ultimately defined by the model itself, and we cannot improve it.

**Reviewer #2:**

**General comments:**

In this paper snow albedo effect is studied in the Europe in winter and spring time. Simulations from Regional Climate Models are used to produce Snow Albedo Sensitivity Index (SASI) and these RCM based SASI values are compared to the SASI values derived from reanalysis and satellite observations. Conclusions are that accurate retrieval of SASI is more dependent of correct snow cover simulations than chosen atmospheric models. This leads to the observation that choosing correct Land Surface Model have an important role in simulating snow albedo effect. The subject itself is very interesting, and this study would be a good fit to The Cryosphere journal. However, I have some concerns.

Thank you very much for your constructive comments, which helped improve the quality of the manuscript. We have followed most of your recommendations as explained in this document and have done a lot of work in the different sections to: 1) clarify the motivation of this work, 2) Better explain the specificities of the different datasets we used, 3) better present our results, and the comparison the inter model differences.

**Specific comments:**

-    I have some difficulties to understand how the satellite based SASI is formed. First, where the SW data is from? Secondly, if snow cover extent from MODIS is used, and that covers years 2003-2015, then I assume that satellite based SASI is covering years 2003-2015. Therefore, satellite based SASI cannot be used to verify/compare RCMs based SASI covering year 1986-2015. Due to the climate change, snow cover extent is vastly different in 2000s than in 1980s or 1990s (e.g. https://www.ncdc.noaa.gov/snow-and-ice/extent/snow-cover/nhland/4). I have doubts that weighting every grid point by the amount of MODIS data is enough to make satellite based SASI comparable to reanalysis and model based SASI.

In the updated manuscript, we omit the weighting of MODIS data and calculate monthly means of MODIS snow cover without applying weights. We tested how calculating monthly means without weighting the data changes MODIS snow cover and found only minor differences (see Figure R2).

[Figure]

Figure R2: Comparison of surface snow cover for weighted and unweighted monthly data from MODIS-TERRA and MODIS-AQUA.

-The value 0.4 as the average albedo difference between a snow-covered and snow-free surfaces is problematic. First, chosen three areas have different vegetation. The Scandinavian area are mostly boreal forest (needleleaved evergreen forest) whereas the East Baltic and the East Europe have more deciduous trees. Throughout the year needleleaved evergreen trees have their "leaves" on, but deciduous trees don't. So difference between snow-covered and snow-free surface albedo should be different in the Scandinavian area (high-latitudes) compared to the two other regions (midlatitudes). Also, the difference depends on whether snow is new or old. Old snow can have impurities, which lowers albedo (Warren and Wiscombe, 1980)). And, in winter snow can sporadically accumulate on trees, which itself increases albedo. I suggest that authors modify âα corresponding to better different scenarios.

Indeed, albedo is influenced by many factors, including forest type and snow age. A constant value of 0.4 was chosen for the change in albedo from afforestation because: 1) the differences between the albedo for snow-covered deciduous forests and evergreen forests is very small. Barlage et al. (2005) show that the values for snow covered evergreen needleleaf forests and deciduous broadleaf forest were 0.34 and 0.35, respectively. Compared to errors elsewhere in the simulations, this difference is very small; And 2) The same value for the change in albedo is used in both the simulations and the observations, so changing this value depending on forest type would have no impact on the comparison between observations and simulations.

**References**:
Barlagem M., Zeng, X., Wei, H., and Mitchell, K.E.: A global 0.05º maximum albedo dataset of snow-covered land based MODIS observations, Geophys. Res. Lett., 32, https://doi.org/10.1029/2005GL022881, 2005.

**Minor comments:** The description of LUCAS experiments will need some clarifications and more details. It is of course allowed to specify manuscript to people with certain scientific knowledge, but as not every reader is familiar with climate models, it would be reader-friendly to provide more explanations.

Thank you for your suggestions to make the manuscript more reader-friendly regarding model features. As explained in the following answers, information has been clarified and/or re-organized for better presenting the LUCAS experiments.

What is a rotated coordinate system, could that term be explained?
In the related location, we added the definition of a rotated coordinate system: "RCMs in LUCAS use a rotated coordinate system**, which is a cartographic projection to transform coordinates from a 3D sphere to a 2D plane (the model domain),** ...".

What is the time resolution of the simulations? Hourly, daily, monthly?
The time resolution at which output have been stored varies from one variable to another and follows the CORDEX protocol https://is-enes-data.github.io/CORDEX_variables_requirement_table.pdf). A sentence has been added to the text to provide information on this.

In line 121 is said that "outputs from ten … RCM simulations", are there more than those chosen ten? If yes, why those specific ten simulations are chosen?
When the study was carried out, among the 11 research teams that had performed LUCAS Phase 1 experiments, ten teams agreed to provide their model output in the requested format (e.g., monthly mean). Meanwhile, other research teams have joined LUCAS. The full model list and model features are reported here: https://docs.google.com/spreadsheets/d/15u5OOHz2NoCxxZZ6OdJyG2akwgSYDqXmV_26xvTeeaY/edit#gid=216121586. We added a sentence in the text to clarify this point, mentioning the availability of the models at the time we performed the analysis.

Table 1, could you open the used acronyms in Table 1 caption?
Table 1 caption has been modified to: "**Table 1:** Summary of participating Regional Climate Models and their Land Surface Models."

Are RCMs WRF 3.8.1 and WRF 3.8.1D the same? If not, what is the difference?

The WRF3.8.1 and WRF3.8.1D model use distinct Planet Boundary Layer schemes as specified at lines 132-134. To make this distinction clearer, we added the following sentence: "... use distinct planetary boundary layer (PBL) schemes**; for this reason, in Table 1 we named differently the two model versions (WRF 3.8.1 and WRF 3.8.1D).**"

Lines 158-168: snow schemes of CLM versions, Noah-MP and RCA4 system are described, but what about iMOVE, VEG3D and TERRA-ML?

Section 2.1.3 has been modified and now includes a description of all the snow schemes:

"In the ensemble, some LSMs contain more sophistication than others. CLM5.0 (Lawrence et al., 2020) and NoahMP (Niu et al., 2007) treats separately canopy-intercepted snow and more realistically captures temperature and wind effects on snow processes. In addition, LSMs differ in the number of additional layers for snow calculation: CLM5.0 uses 12 snow layers; CLM4.0, CLM4.5 and TERRA-ML (Tolle et al., 2018) use five, three in NoahMP, and two in VEG3D and iMOVE, and one in the RCA4 model. The iMOVE model adopts the snow parameterisation from the global climate model ECHAM4 (Roeckner, et al., 1996) and reproduces the snow albedo as a linear function of the snow surface temperature and of the forest fraction in a grid cell, with fixed maximum and minimum snow albedo at temperatures lower than -10°C and at 0°C, respectively (Kotlarskis, 2007). In the VEG3D model, the snow scheme is based on the Canadian Land Surface Scheme (CLASS) (Verseghy, 1991) and ISBA (Douville et al., 1995) and accounts for changes of surface albedo and emissivity as well as processes like compaction, destructive metamorphosis, the melting of snow, and the freezing of liquid water."

Lines 180-181: the two thresholds for cloud cover are used. 50% of cloud cover is quite a lot of clouds in one cell, why this threshold was chosen? How were these thresholds used?
In the main part of the paper we only show results using a threshold of 50%. This is now clarified in the text. To examine how sensitive the results are to the choice of thresholds, we also added the snow cover fraction obtained by selecting a threshold of 20% to the MODIS data in Supplementary Figure S1. The results are very similar but with a slight tendency of higher snow cover fractions when using a threshold of 20%.

Line 182: why also "good" and "ok" flagged data was used?
We assumed that the quality flags "good" and "ok" still deliver reliable data. To check whether the inclusion of these data makes a large difference, we made an alternative version of Figure 5, only using the MODIS data flagged "best" (Figure R3). The results are very similar to Figure 5. Some differences in the month January (and the month February in Scandinavia) are likely due to the reduced sample size when only using the flag "best" (note that for January no data in Scandinavia are left in this case). We thus keep "good" and "ok" flagged data in Figure 5.

[Figure]

Figure R3: As Figure 5 but only using data that are flagged as "best".

Line 209: mention that the Scandinavian region have mostly needleleaved evergreen forest, whereas other two regions have more deciduous trees.
Thank you, this point has been clarified in the text.

line 277: The peaks are quite pronounced in the East Europe and the East Baltic regions, but I think they are less pronounced in the Scandinavian region. Could it be due to the illumination conditions?
Thank you for the suggestion, looking at Figures 6 and S5, showing the incoming SW radiation, we can see that East Baltic and East Europe present generally higher values than Scandinavia. So, the illumination conditions could be one of the factors explaining the difference between Scandinavia and the other regions. This point will be clarified in the text.

line 312: based on Figure 5, the snow cover for MODIS is from MODIS-TERRA, is that correct? Why MODIS-TERRA, if you also have MODIS-AQUA data?
MODIS-TERRA and MODIS-AQUA are not included in the figure.
As we are omitting the weighting of MODIS data in the updated version of the manuscript, we now show both MODIS-AQUA and MODIS-TERRA in the supplemental material and only MODI that include snow cover data.

line 314: what are those limitations and biases that are referenced to in this sentence?
Thank you for mentioning this point, we now give an example for the reanalysis and also cite an article presenting some of the biases in reanalysis and satellite observations for snow.

Daloz, A. S., Mateling, M., L'Ecuyer, T., Kulie, M., Wood, N. B., Durand, M., Wrzesien, M., Stjern, C. W., and Dimri, A. P.: How much snow falls in the world's mountains? A first look at mountain snowfall estimates in A-train observations and reanalyses, The Cryosphere, 14, 3195–3207, https://doi.org/10.5194/tc-14-3195-2020, 2020.

line 328: also WRFb-CLM4.0 have high values during the ablation period. Should that model also be added?
Although this is true for East Europe, WRb-CLM4.0 does not show such high values in SASI over the two other regions compared to the other simulations, so we prefer focusing on RegCMb-CLM4.5 and CCLM-VEG3D in this part of the article.

lines 322-334: would it be more informative to add different markers whether models are over or under the range of reference datasets? For example, black dots when over and red?) x when under?
Thank you for the suggestion, but we decided to not introduce extra colors in the figures, since it already contains a lot of information.
line 346- 349: I would argue that REMO-iMOVE and WRFa-NoahMP have very different results, not REMO-iMOVE and CCLM-VEG3D, if these results are based on Figure 5. But also, based on Figure 3, CCLM-VEG3D do not reproduce SASI well at all.
Thank you, this is indeed a mistake that has been corrected in the text.
Figure2: colorbar ticks and color limits do not match, could it be modified?
Figure 2, which is now Figure 3, has been modified, colorbar ticks and color limits should match in this new version.
Figure 3 and 4: can horizontal lines be added? It would make reading of the figures much easier. Also, it would be more informative to draw ERA5 and SATELLITE lines last so they would be top of everything.
Figure 4 has been removed while Figure 3 has been modified with ERA5 and Satellite on top, as suggested.
Figure 5: black color of MODIS-TERRA, especially black median line in the very dark grey bar is difficult to see. Can bar be made more lighter grey?
We adjusted the color scheme of the reanalyses and satellite observations, so that the median is easier to see.
Table 1: Can table rows be listed based on RCM (as in Figure 2), not institute? It would be easier to read.
Thank you for the suggestion, Table 1 has been modified, it is indeed easier to read it.

**Technical corrections:**
lines 46-47: word "it" is ambiguous, could this sentence be modified to be easier to read?
This sentence has been modified to avoid the ambiguity.
line 80: open the "RCMs" acronym
This now corrected in the text.
line 107: "Section4 the last sections" -> remove "the last section"
This now corrected in the text.
line 175: add MODIS product names (for TERRA: MOD10C1 and for AQUA: MYD10C1)
We have added the names of the MODIS product names.
line 176: reference to the same section 2.2 is not necessary, remove it or change it

This reference has been removed from the text.
line 225: "The model data.." -> change it to "Most of the model data …"
This modification has been included in the text.
line 319: ".. snow cover varies between…" -> "..snow cover mean varies between…"
This is now included in the new version of the manuscript.

References:
Warren, S. G., & Wiscombe, W. J. (1980). A Model for the Spectral Albedo of Snow. II: Snow Containing Atmospheric Aerosols, Journal of Atmospheric Sciences, 37(12), 2734-2745. Retrieved Nov 15, 2021, from https://journals.ametsoc.org/view/journals/atsc/37/12/1520-0469_1980_037_2734_amftsa_2_0_co_2.xml